# Multi-layered heterochromatin interaction as a switch for DIM2-mediated DNA methylation

Zengyu Shao [1], Jiuwei Lu [1], Nelli Khudaverdyan[1] & Jikui Song [1] ✉

Functional crosstalk between DNA methylation, histone H3 lysine-9 tri-methylation (H3K9me3) and heterochromatin protein 1 (HP1) is essential for proper heterochromatin assembly and genome stability. However, how repressive chromatin cues guide DNA methyltransferases for region-specific DNA methylation remains largely unknown. Here, we report structure-function characterizations of DNA methyltransferase Defective-In-Methylation-2 (DIM2) in *Neurospora*. The DNA methylation activity of DIM2 requires the presence of both H3K9me3 and HP1. Our structural study reveals a bipartite DIM2-HP1 interaction, leading to a disorder-to-order transition of the DIM2 target-recognition domain that is essential for substrate binding. Furthermore, the structure of DIM2-HP1-H3K9me3-DNA complex reveals a substrate-binding mechanism distinct from that for its mammalian orthologue DNMT1. In addition, the dual recognition of H3K9me3 peptide by the DIM2 RFTS and BAH1 domains allosterically impacts the DIM2-substrate binding, thereby controlling DIM2-mediated DNA methylation. Together, this study uncovers how multiple heterochromatin factors coordinately orchestrate an activity-switching mechanism for region-specific DNA methylation.

In eukaryotes, compartmentalization of chromatin into transcription-active euchromatin and repressive heterochromatin is essential for gene regulation and genome stability[1]. Establishment and maintenance of proper heterochromatin assembly depends on an intertwined network of epigenetic mechanisms, such as histone H3 lysine 9 tri-methylation (H3K9me3) and DNA methylation at the C-5 position of cytosine[2,3]. Specific readout of these epigenetic marks by respective reader proteins, such as heterochromatin protein 1 (HP1) for H3K9me3[4–6] and methyl-cytosine binding proteins for DNA methylation[7], leads to formation of a compact heterochromatin environment underlying gene silencing[8]. Mechanistic understanding of the functional interplay between these epigenetic pathways is crucial for deciphering the mechanisms behind gene regulation and cell fate determination.

C-5 cytosine DNA methylation is widely present in eukaryotic species, ranging from mammals, plants to filamentous fungi[9,10]. However, the DNA methylation machinery and its substrate specificity have diversified throughout evolution. In mammals, DNA methylation predominantly occurs within the CG dinucleotide context, accounting for ~70% of total CG sites[11]. Mammalian DNA methylation is mainly established by de novo DNA methyltransferases DNMT3A and DNMT3B, together with a non-catalytic paralog DNMT3L, and maintained by DNA methyltransferase 1 (DNMT1)[12–15]. Plant DNA methylation occurs in all sequence contexts: CG, CHG, and CHH (H = A, C, T)[14], all of which are established by DOMAINS REARRANGED METHYL-TRANSFERASE 2 (DRM2)[14]. In *Arabidopsis*, DNA METHYLTRANSFERASE 1 (MET1) mediates the maintenance of CG methylation[16], CHROMO-METHYLASE 3 (CMT3) mediates the maintenance of CHG methylation[17], while CHROMOMETHYLASE 2 (CMT2) and DRM2 mediate CHH methylation at long heterochromatic transposable elements (TEs)[18] and short euchromatic TEs, respectively[19]. In *Neurospora*, DNA methylation accounts for 1.5% of total cytosines in all sequence contexts[20–22], with Defective In Methylation-2 (DIM2) as the sole DNA methyltransferase in vegetative cells[23]. It has been established that

[1]Department of Biochemistry, University of California, Riverside, CA 92521, USA. ✉e-mail: jikui.song@ucr.edu

DNA methylation in *Neurospora* is mostly enriched in the constitutive heterochromatin, closely associated with the AT-rich relics of repeat-induced point mutation (RIP)[24,25]. However, due to lack of mechanistic knowledge, how DIM2 orchestrates such unique yet complexed methylome in *Neurospora* is unknown.

Functional crosstalk between DNA methylation and repressive histone modifications is an evolutionarily conserved mechanism that underpins stable maintenance of heterochromatic assembly[26]. Both mammalian DNMT1 and plant CMT3 harbor reader modules that specifically recognize histone H3K9 methylation, which in turn allosterically regulates DNMT1- and CMT3-mediated DNA methylation[27–30]. Likewise, DIM2-mediated DNA methylation in *Neurospora* was shown to be initiated by Defective-In-Methylation-5 (DIM-5), a histone H3K9me3 writer[31]. The mechanistic link between DIM2-mediated DNA methylation and heterochromatin is further strengthened by the observation of a direct interaction between DIM2 and HP1[32]. Mutation of either DIM-5 or HP1 led to DNA methylation abolishment in *Neurospora*[31,33,34], highlighting an essential role of heterochromatin regulation in DIM2 activity. To date, the molecular basis for the functional regulation of DIM2 by HP1 and H3K9me3 remains elusive.

DIM2 belongs to the superfamily of DNMT1, bearing ~16% sequence identity with human DNMT1[21,23]. Like DNMT1, it is predicted to contain a C-terminal methyltransferase (MTase) domain preceded by an N-terminal tail (NT), a putative replication-foci-targeting sequence (RFTS) domain and a pair of bromo-adjacent homology (BAH1 and BAH2) domains (Fig. 1a)[23]. The corresponding RFTS, BAH1 and MTase domains in DNMT1 have been characterized previously[15,29,35–40]. For instance, the DNMT1 MTase domain strictly methylates CG sites, with marked substrate preference for hemimethylated CGs[13,37,41]; the DNMT1 RFTS domain interacts with the MTase domain to inhibit the DNA binding of DNMT1[38–40], which can be relieved through the interaction between the RFTS domain and histone H3K9me3 and H3 mono-ubiquitylated at lysine 14, 18 and/or 23[29,42–45]; and the DNMT1 BAH1 domain serves as a specific reader for histone H4 lysine 20 trimethylation (H4K20me3) to fine-tune the region-specific DNA methylation in L1 transposons[35]. In contrast, the biochemical roles of the functional domains of DIM2 have yet to be characterized. Considering that *Neurospora* and mammalian genomes are associated with distinct patterns of cytosine methylation (all sequence contexts in *Neurospora* vs. CG-specific in mammals)[21], understanding the structure and mechanism of DIM2-mediated DNA methylation is important for unraveling how DNA methyltransferases have evolved to adapt various genomic complexities.

To delineate the mechanistic basis of DIM2-mediated DNA methylation, we set out to investigate the functional regulation of *Neurospora Crassa* DIM2 by HP1 and H3K9me3. Unlike most other DNA methyltransferases characterized so far that, in the absence of regulatory proteins, retain a basal DNA methylation activity[18,27,46–50], DIM2 requires the presence of HP1 and H3K9me3 for a notable DNA activity. Next, we solved the single-particle cryogenic electron microscopy (cryo-EM) structures of DIM2 in complex with HP1, DIM2 in apo form (apo-DIM2), and DIM2 in complex with HP1, H3K9me3 peptides and a DNA duplex. Structural and biochemical analysis of the DIM2-HP1 complex reveals a bipartite interaction between DIM2 and HP1, involving the N-terminal tail (NT) and the target recognition domain (TRD) of DIM2 and the chromo shadow domains (CSDs) of HP1 dimer. In comparison with apo-DIM2, the interaction between DIM2 TRD and HP1 CSDs leads to a disorder-to-order transition of the TRD, providing an explanation for the requirement of HP1 in DIM2-mediated DNA methylation. Furthermore, the structure of DIM2 in complex with HP1, H3K9me3, and DNA reveals that the DIM2 RFTS domain, unlike its counterpart in DNMT1 that functions as a DNA-competitive inhibitor[38–40], forms an additional DNA-binding site to facilitate substrate association. The interaction between DIM2 and H3K9me3 peptides is mediated by both the RFTS and BAH1 domains of DIM2, with

the RFTS-H3K9me3 interaction positioning H3 K14 for a direct interaction with substrate DNA, and the BAH1-H3K9me3 interaction repositioning a loop bridging the TRD and the catalytic core of DIM2 for DNA contact. Disruption of the DIM2 residues key for HP1, H3K9me3, or DNA binding severely impairs the DNA methylation activity of DIM2. Together, this study unravels a regulatory model for DIM2, where multiple interactions of heterochromatin factors synergistically control DIM2-mediated DNA methylation, with important implications in the functional crosstalk between DNA methylation and heterochromatin cues.

## Results

### HP1 and H3K9me3 are both required for DIM2-mediated DNA methylation

To elucidate how DIM2-mediated DNA methylation interplays with heterochromatin factors HP1 and H3K9me3, we performed in vitro DNA methylation assay for an N-terminal fragment of DIM2 (residues 1-1242), spanning the NT, RFTS, BAH1, BAH2, and the MTase domains (Fig. 1a), on a 36-mer DNA duplex containing multiple cytosines (Fig. 1b). Surprisingly, under the experimental condition, DIM2 alone showed no detectable activity (Fig. 1b). The presence of histone H3 peptide (residues 1-24) harboring the H3K9me3 modification ($H3_{1-24}$K9me3) failed to activate DIM2 appreciably, while the presence of full-length HP1 led to a slight but notable activity of DIM2 (Fig. 1b). In contrast, the presence of both HP1 and H3K9me3 peptide greatly increased the DNA methylation efficiency of DIM2, higher than that of DIM2-HP1 mixture by ~90-fold (Fig. 1b and Supplementary Fig. 1a), suggesting a synergistic activity-stimulating effect between HP1 and H3K9me3. Note that such an activity-stimulation effect appears specific to heterochromatin factors, as the replacement of the H3K9me3 peptide by the H3 peptide containing unmethylated H3 K9 (H3K9me0) failed to synergize with HP1 to boost the DNA methylation activity of DIM2 (Fig. 1b). The requirement for HP1 protein is unlikely related to its H3K9me3-binding activity, as our Isothermal Titration Calorimetry (ITC) analysis revealed that impairing the H3K9me3-binding activity of HP1 via the H3K9me3-cage mutation W98A[51] (Supplementary Fig. 1b−d and Supplementary Table 1) failed to abolish HP1-mediated regulation; rather, it increases the activity-stimulation effect of HP1 on DIM2 by ~2-fold (Supplementary Fig. 1e).

Next, we compared the DNA methylation activity of DIM2 on DNA substrates containing one central CG, CHG or CHH target site. As expected, DIM2 methylates both CG and non-CG DNAs (Supplementary Fig. 1f). Furthermore, under the experimental condition, DIM2 shows a modest substrate preference for hemimethylated over unmodified CG DNA. The caveat of this observation is that only one set of CG, CHG and CHH substrates were used. A detailed understanding of how the DNA sequence context impacts the activity of DIM2 awaits further investigation.

### Structural overview of the DIM2-HP1 complex

Consistent with a previous report that DIM2 directly interacts with HP1[32], our size-exclusion chromatography analysis reveals comigration of DIM2 (residues 1-1242) fragment and full-length HP1 (Supplementary Fig. 1g), confirming the formation of the DIM2-HP1 complex. Subsequently, we solved the cryo-EM structure of DIM2 (residues 1-1242) in complex with HP1 and cofactor byproduct S-adenosyl homocysteine (SAH) (denoted as DIM2-HP1 complex thereafter) at overall 2.76-Å resolution (Fig. 1c−f, Supplementary Fig. 2 and Supplementary Table 2). Structural analysis of the DIM2-HP1 complex reveals that one DIM2 molecule binds to a homodimer of HP1 CSD domain (HP1-I and HP1-II) (Fig. 1c−f). We were able to trace the majority of DIM2 residues, except for M1-Q125, D436-E546, D562-D578, K592-S602, K699-T702, T933-K936, K1143-D1148 (Fig. 1c, d and Supplementary Fig. 3a−e). On the other hand, although full-length HP1 was used for cryo-EM study, we were only able to trace the density for the

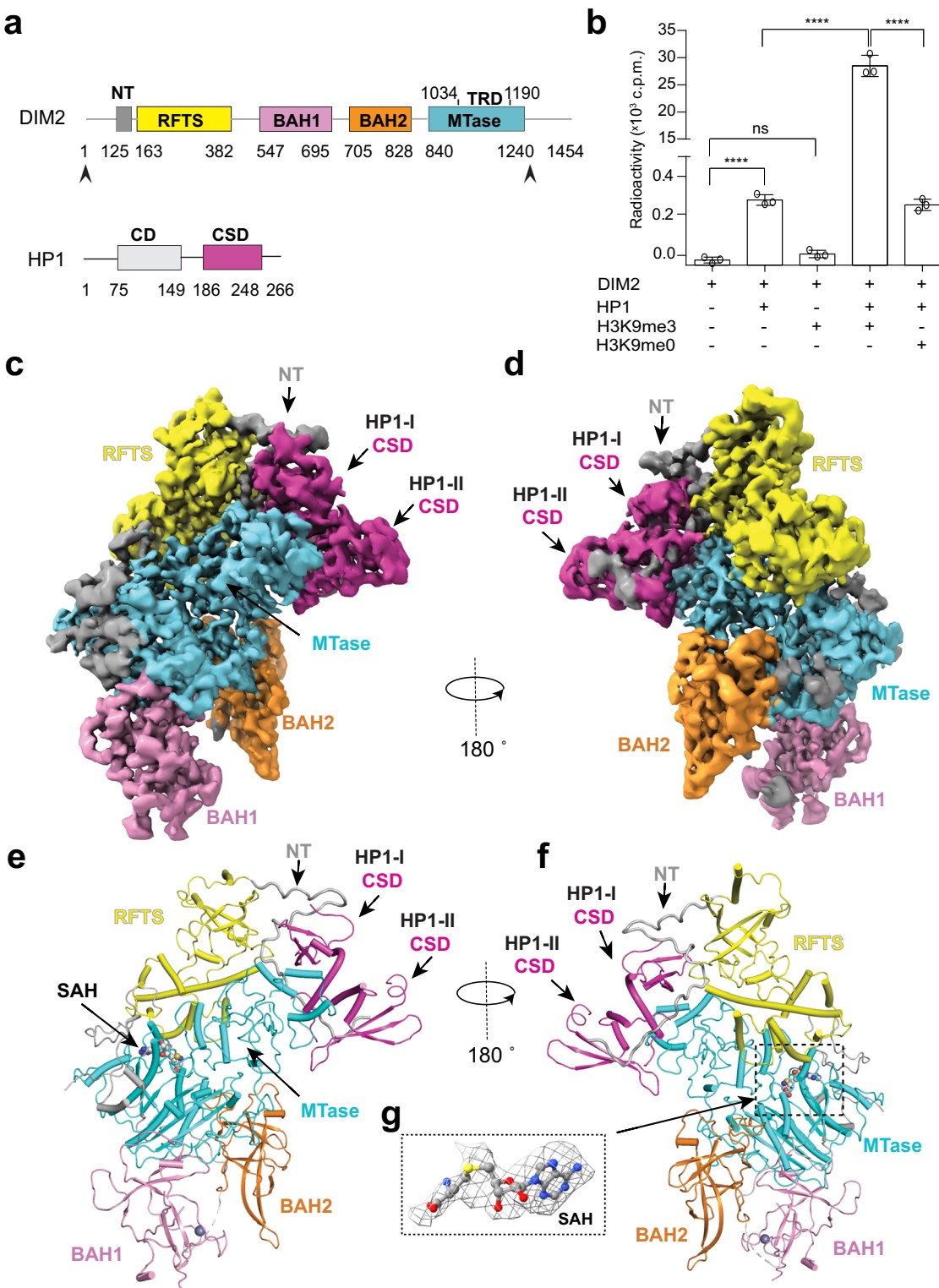

**Fig. 1 | Cryo-EM structure of the DIM2-HP1 complex. a** Domain architecture of DIM2 and HP1, with individual domains color coded and delimited by amino acid numbers. The DIM2 fragment used for structural and biochemical characterization is indicated by arrows. **b** In vitro DNA methylation assay of DIM2 on a $(CTA)_{12}$/ $(TAG)_{12}$ 36-mer DNA duplex, in the presence or absence of HP1 and/or histone peptides (H3$_{1-24}$K9me3 or H3$_{1-24}$K9me0). Data are mean ± s.d. ($n$ = 3 biological replicates). The two-tailed Student's $t$ test statistical analysis was performed to compare DIM2 activity in the absence vs. presence of HP1 and/or histone peptides.

ns not significant; ****$p$ < 0.0001. Source data are provided as a Source Data file. **c**, **d** Two oppositive views of the cryo-EM density map of the DIM2-HP1 complex. **e**, **f** Atomic model of the DIM2-HP1 complex in two opposite views, with individual domains of DIM2 color coded. The two HP1 CSD subunits are colored in magenta. The NT and domain linkers are colored gray. The SAH molecule and zinc ion are shown in sphere representation. The color scheme for the individual domains of DIM2 is applied to subsequent figures unless otherwise indicated. **g** Cryo-EM density map for the SAH molecule bound to DIM2.

CSD domains (residues K189-I251 of HP1-I CSD and residues C191-I251 of HP1-II CSD) (Supplementary Fig. 3f).

The structure of DIM2 reveals that the N-terminal domains, composed of the extended NT and the RFTS, BAH1 and BAH2 domains, are centered around the C-terminal MTase domain (Fig. 1c–f): the RFTS domain is positioned on one side, and the BAH1 and BAH2 domains are positioned on the other side, reminiscent of what was observed for DNMT1[39,40]. The RFTS domain is followed by an extended linker that traverses the catalytic core, forming a pair of α-helices anchored next to the catalytic site (Fig. 1e, f). The MTase domain is further comprised of two subdomains: the catalytic core and the TRD, with the catalytic core harboring the active site and the SAH molecule (Fig. 1e–g and Supplementary Fig. 3g). On the other hand, the structure of HP1 CSD dimer resembles that of mammalian HP1[51], with each CSD containing a three-stranded antiparallel β-sheet that is packed by two C-terminal α-helices (Fig. 1e, f).

### Structural basis for the DIM2-HP1 interaction

Association of DIM2 with HP1 CSDs is mediated by a bipartite interaction, involving one interface formed by the canonical peptide-binding groove of the CSD homodimer[51,52] and a portion of the NT and TRD of DIM2 (Interface I), and another interface formed by the C-terminal helix-proximate region of HP1-I CSD and DIM2 TRD (Interface II) (Fig. 2a–d).

At the interface I, the N-terminal region (residues Q125-H129) of the DIM2 NT pairs in parallel with the first β-strand of HP1-II CSD via hydrogen-bonding interactions between DIM2 S127 and HP1 H247′ (prime symbol denotes residues from the HP1-II) and between DIM2 H129 and HP1 A204′. Next, the DIM2 NT diverts by 90° at residue H129 and runs in an extended form along the surface groove formed by the C-terminal tails of both HP1 CSDs (Fig. 2a, c), mediated by backbone hydrogen bonds involving DIM2 T131, V132, D133 and L134 and HP1 H247, R249 and I251. In addition, residues I130, V132, L134, and P135 of DIM2 engage in non-polar contacts with HP1 (L213, F243, Y244, and V248 of HP1-I and residues F243′, Y244′, V248′, I250′, and I251′ of HP1-II), and residue D133 of DIM2 forms a salt bridge with residue R249′ of HP1-II (Fig. 2c). Note that the positioning of residues I130, V132, and L134 of DIM2 matches the −2, 0, and +2 sites of the canonical HP1-interacting $P_{-2}XV_0XL_{+2}$ motif[51,52], respectively, indicative of a canonical interaction mechanism. At the exit of the surface groove, DIM2 L139 further engages in a hydrophobic contact with HP1-I Y216, and DIM2 T138 forms a hydrogen bond with HP1-I A204 (Fig. 2c). Subsequently, residues N141-R148 of DIM2 divert by 90° at residue N141 to traverse the β-sheet of the HP1-I CSD, involving a salt-bridge interaction between DIM2 R143 and HP1-I D207, and hydrogen-bonding interactions between DIM2 R143 and Q147 and HP1-I Y216 and K224, respectively (Fig. 2c). Additional intermolecular interactions at the interface I include a salt bridge between DIM2 R1104 and HPI-I D203, and hydrogen-bonding interactions involving DIM2 A1108 and D1185 and HP1-I Q242 and R246 (Fig. 2c, e).

At the interface II, the helical segments (residues 1101-1138) of DIM2 TRD are packed against the β-sheet HP1-I CSD as well as the loop bridging the first two β-strands of HP1-II CSD (Fig. 2d), engaging in intermolecular hydrogen-bonding interactions involving DIM2 H1113, Y1115 and D1134, HP1-I K239, and V248, and HP1-II T210′, and van der Waals contacts involving DIM2 H1113, P1114 and Y1115, and HP1-II H211′ and K212′ (Fig. 2d). Together, these interactions underpin the association between DIM2 and HP1 CSD homodimer.

To test the structural observations, we selected key HP1-interacting residues (L134, L139 and R1104) (Fig. 2c, e) of DIM2 for mutagenesis, followed by size-exclusion chromatography and in vitro DNA methylation analysis. Introducing the L134A/L139A double mutation to DIM2 led to partial disruption of the DIM2-HP1 complex, as indicated by the presence of three discrete elution peaks corresponding to the DIM2-HP1 complex, DIM2 and HP1, respectively (Fig. 2f

and Supplementary Fig. 4a–d). Introducing the R1104A mutation did not affect the assembly of the DIM2-HP1 complex appreciably (Supplementary Fig. 4e). However, introducing the L134A/L139A/R1104A triple mutation to DIM2 (DIM2^LLR) led to complete disruption of the DIM2-HP1 complex, as indicated by the disappearance of the elution peak for the DIM2-HP1 complex (Fig. 2f and Supplementary Fig. 4f). Consistently, our in vitro DNA methylation assay showed that introducing the L134A/L139A double mutation reduced the DNA methylation efficiency of DIM2 by 14-folds, whereas introducing the DIM2^LLR mutation completely abolished the DNA methylation activity of DIM2 (Fig. 2g). These data lend a strong support for the observed DIM2-HP1 interaction and its critical role in DIM2-mediated DNA methylation.

### Mechanistic basis for HP1-mediated allosteric regulation of DIM2

To illustrate the mechanism by which HP1 binding controls DIM2-mediated DNA methylation, we further solved the cryo-EM structure of apo-DIM2 at 2.88-Å resolution (Fig. 3a–d and Supplementary Figs. 5, 6 and Supplementary Table 2). We were again able to trace the majority of DIM2 molecule, except for residues M1-Q160, D436-K541, D562-D578, K592-N601, L931-K936, and S1097-G1152 (Fig. 3a–d and Supplementary Fig. 6a–e). As with the DIM2-HP1 complex, the MTase domain of apo-DIM2 harbors a SAH molecule in the active site (Fig. 3e and Supplementary Fig. 6b). Structural alignment of apo-DIM2 with HP1-bound DIM2 gave a root-mean-square deviation (RMSD) of 1.12 Å over 873 aligned Cα atoms (Fig. 3f), indicative of structural similarity. Nevertheless, substantial structural changes were observed for the HP1-interacting regions: residues Q125-S161 of the NT and residues A1096-W1153 of the TRD of DIM2 are well defined in the DIM2-HP1 complex but become disordered in apo-DIM2 (Fig. 3g), suggesting that the HP1 binding induces a disorder-to-order transition of the DIM2 NT and TRD. In addition, less pronounced but notable conformational changes were observed for residues N198-Q204 and A213-K220 of the RFTS domain and residues S1172-G1179 of the TRD (Fig. 3g), reflecting an indirect effect of HP1 interaction.

The TRD of a DNA methyltransferase is knowingly essential for its substrate binding[15,53]. The fact that the DIM2 residues A1096-W1153, which account for over one-third of the DIM2 TRD (Fig. 1a), undergo a disorder-to-order transition upon HP1 binding raises a possibility that HP1 binding allosterically activates DIM2 through structural stabilization of the DIM2 TRD. To test this notion, we first performed thermal shift assays to compare the stabilities of wild-type (WT) DIM2 with that harboring a TRD mutation, in which the HP1-interacting residue R1104 was replaced by an alanine (R1104A). WT and R1104A DIM2 both yielded a melting temperature (Tm) of 39.5 °C (Supplementary Fig. 7a, b), in line with the fact that the TRD is structurally disordered in apo-DIM2. Co-incubation of WT DIM2 with HP1 led to a 5.0 °C increase of Tm (Supplementary Fig. 7a, b), consistent with the formation of DIM2-HP1 complex that further stabilizes the DIM2 TRD. In contrast, co-incubation of R1104A DIM2 with HP1 only increased the Tm by 2.5 °C (Supplementary Fig. 7a, b), supporting the notion that the R1104A, through disruption of the interaction between DIM2 TRD and HP1, reduces the stability of the TRD in the DIM2-HP1 complex. Next, we compared the in vitro DNA methylation activities of WT and R1104A-mutated DIM2. Despite that the DIM2 R1104A mutation alone barely impacts the association of the DIM2-HP1 complex (Supplementary Fig. 4e), the activity-stimulation effect of HP1 on R1104A-mutated DIM2 was reduced by ~8-fold when compared with that on WT DIM2 (Fig. 3h). These observations therefore support the notion that HP1 allosterically regulates the DNA methylation activity of DIM2 via structural stabilization of the DIM2 TRD.

### Structural overview of the DIM2-HP1-H3K9me3-DNA complex

To elaborate how HP1 and H3K9me3 cooperate in activating DIM2-mediated DNA methylation, we further solved the cryo-EM structure of

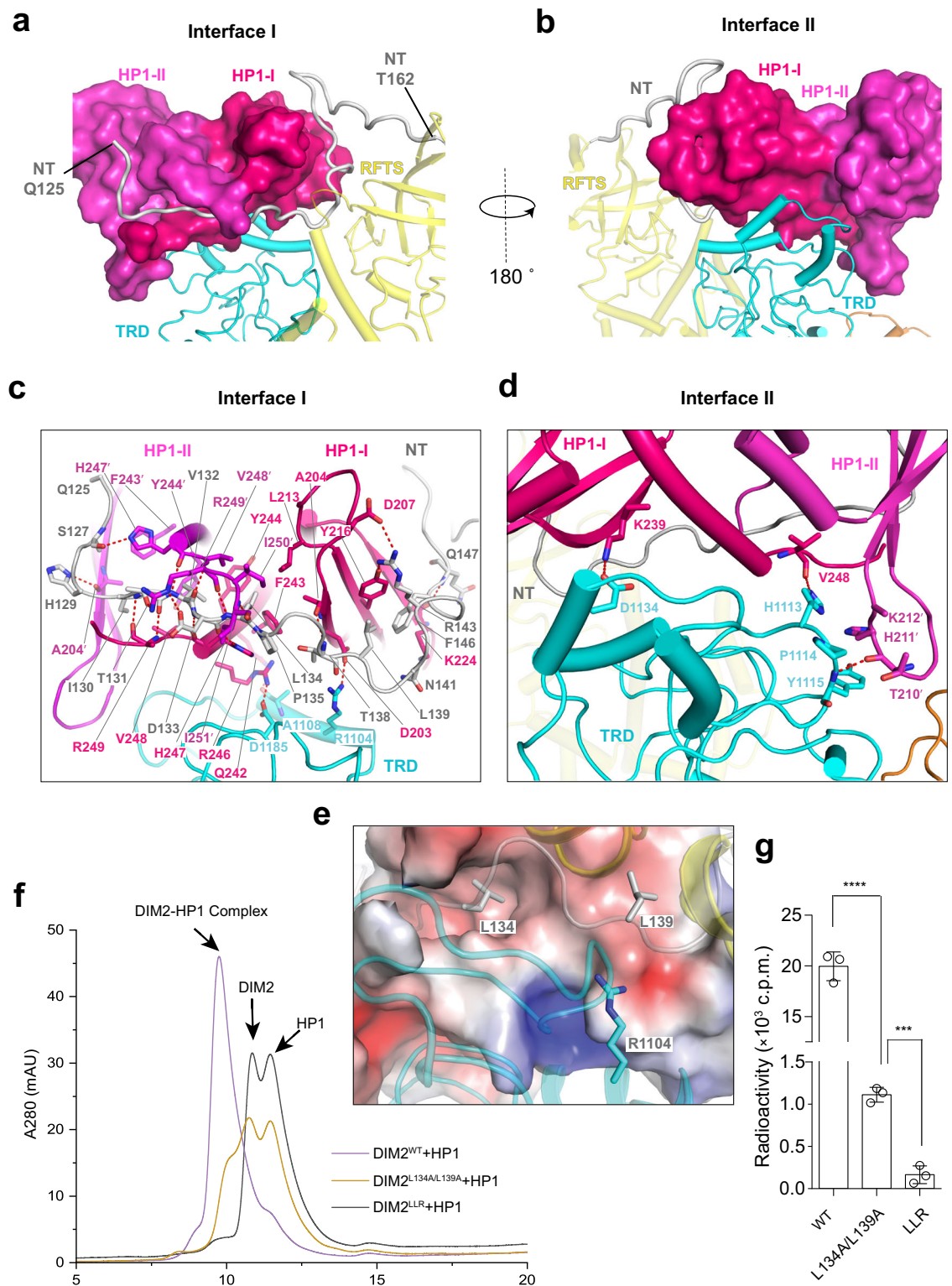

**Fig. 2 | Structural basis for the DIM2-HP1 interaction. a**, **b** Close-up view of the interaction between DIM2 and HP1 homodimer, with the two HP1 CSD subunits shown in surface view (HP1-I: hot pink, HP1-II: magenta) and DIM2 shown in ribbon representation. **c** Close-up view of the interaction between HP1 and DIM2 at interface I. **d** Close-up view of the interaction between HP1 and DIM2 at interface II. **e** Close-up view of key HP1-contacting residues of DIM2, with HP1 shown in electrostatic surface representation. **f** Size-exclusion chromatography analysis of DIM2, WT or mutant, mixed with HP1. **g** In vitro DNA methylation assay of DIM2, WT or mutant, on a $(CTA)_{12}/(TAG)_{12}$ DNA duplex, in the presence of HP1 and histone H3$_{1-24}$K9me3 peptide. LLR, DIM2 L134A/L139A/R1104A triple mutation. Data are mean ± s.d. ($n = 3$ biological replicates). Statistical analysis used two-tailed Student's $t$ test. ***$p = 0.0003$; ****$p < 0.0001$. Source data are provided as a Source Data file.

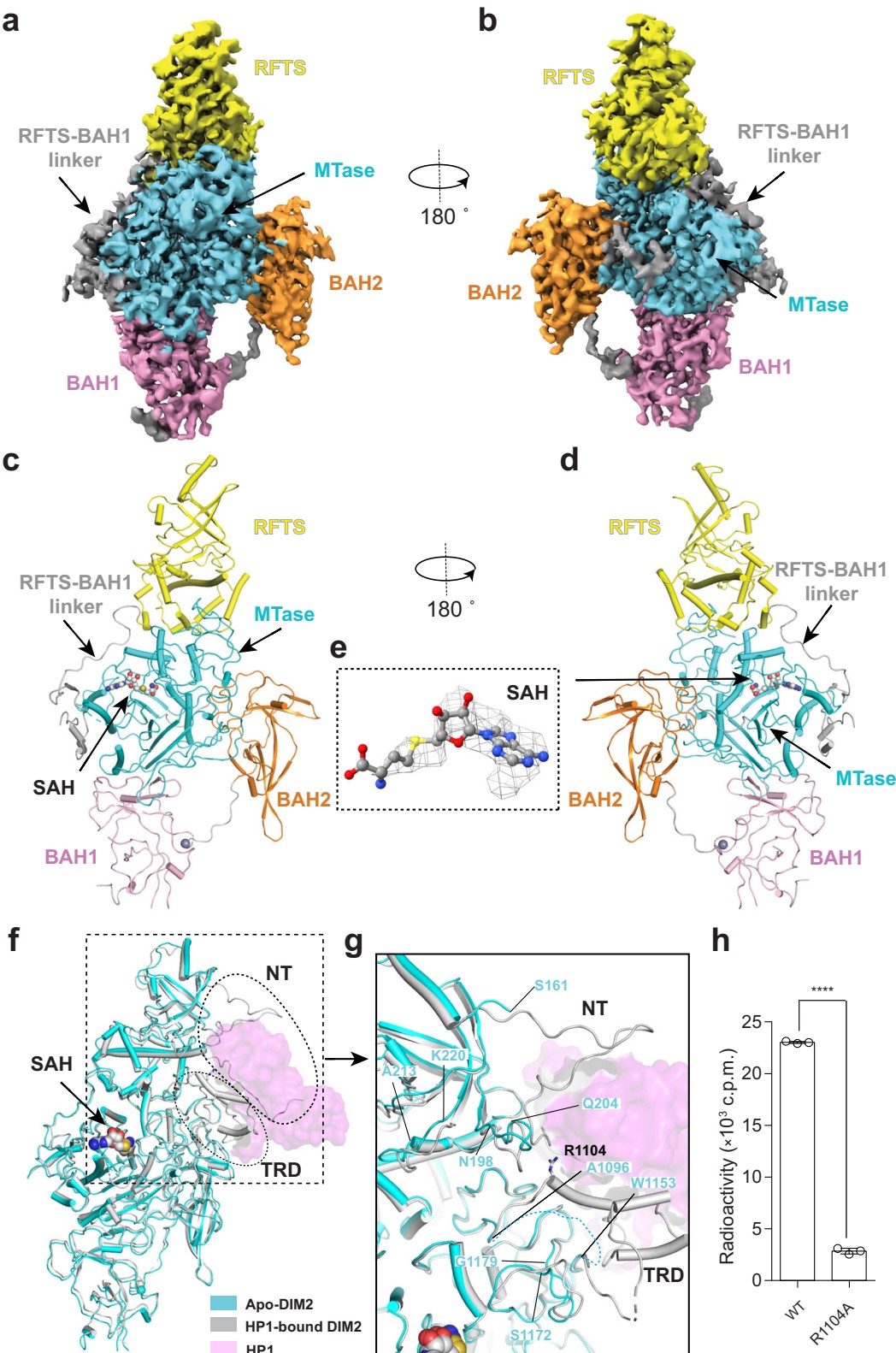

**Fig. 3 | Cryo-EM structure of apo-DIM2. a, b** Cryo-EM density map for apo-DIM2 in two opposite views. **c, d** Atomic model of apo-DIM2 in two opposite views, with individual domains color coded. The SAH molecule and zinc ion are shown in sphere representation. **e** Cryo-EM density map for the SAH molecule bound to DIM2. **f** Structural alignment between HP1-bound DIM2 (gray) and apo-DIM2 (cyan). The NT and TRD segments that undergo a disorder-to-order transition upon HP1 binding are indicated by dotted circles. The SAH molecules and zinc ions are shown in sphere representation. **g** Close-up view of the NT and TRD, with regions that undergo HP1 binding-induced conformational changes are delimited by residue numbers. **h** In vitro DNA methylation assay of DIM2, WT or mutant, on a $(CTA)_{12}/(TAG)_{12}$ DNA duplex, in the presence of HP1 and histone $H3_{1-24}K9me3$ peptide. Data are mean ± s.d. ($n = 3$ biological replicates). Statistical analysis used two-tailed Student's $t$ test. ****$p < 0.0001$. Source data are provided as a Source Data file.

DIM2 in complex with HP1, H3$_{1-24}$K9me3 peptides and an 18-mer DNA duplex. The DNA molecule contains a zebularine (Z7) on the target DNA strand (Fig. 4a), which permits the formation of a stable, productive complex between a DNA methyltransferase and DNA, as demonstrated previously[54–57]. The structure of the DIM2-HP1-H3K9me3-DNA complex was solved at an overall resolution of 2.79 Å (Fig. 4b–d, Supplementary Figs. 8, 9 and Supplementary Table 2).

We were able to trace the entire DNA molecule and most of the DIM2 molecule, except for residues M1-S127, E438-V540, and K592-N601 (Fig. 4b–d and Supplementary Fig. 9a–g). The structure reveals that DIM2 binds to HP1, H3$_{1-24}$K9me3 peptide and DNA molecule in a stoichiometry of 1:2:2:1 (Fig. 4b, d). The chromodomains of HP1 remain untraceable, whereas the homodimeric CSD domains associate with DIM2 in the same manner as that in the DIM2-HP1 complex (Fig. 4d vs. Fig. 1e, f). Notably, the DNA molecule is embedded in the cleft between the catalytic core and TRD (Fig. 4d–f), with the Z7 breaking away from its Watson-Crick pair Gua7' (' denotes the base on the non-target strand) and inserting into the active site for a covalent-linkage with the catalytic cystine C926 of DIM2, where Z7 is further locked in place through interactions with other catalytic residues, such as S924, P925, E966, R1013 and R1015 (Fig. 4e). Each DIM2 molecule binds to two H3K9me3 peptides, with one associated with the RFTS domain (Fig. 4d, g) and the other associated with the BAH1 domain (Fig. 4d, h). We were able to trace residues T6-R17 for the H3K9me3 peptide bound to the RFTS domain (Fig. 4f, g), and residues T3'-G13' (prime symbol denotes residues from the BAH1-bound H3K9me3 peptide) for the one bound to the BAH1 domain (Fig. 4f, h).

Structural overlay of the DIM2-HP1-H3K9me3-DNA complex with the DIM2-HP1 complex reveals high similarity, with an RMSD of 0.65 Å over 809 aligned Cα atoms. Nevertheless, DNA and H3K9me3 peptides-induced conformational changes were observed for residues I209-Y221 of the RFTS domain, residues V401-V408 of the RFTS-BAH1 linker, the catalytic loop (residues S924-V938), and residues R1039-S1053 and G1142-A1149 of the TRD (Supplementary Fig. 10a–c). In addition, a loop in the BAH1 domain (residues R561-H579, denoted as allosteric loop herein) undergoes a disorder-to-order transition upon H3K9me3 binding (Supplementary Fig. 10c).

## Structural basis for the DIM2-DNA substrate interaction

The interaction between DIM2 and the 18-mer DNA spans 16 base pairs and discrete regions of the catalytic core, the TRD, and the RFTS domain of DIM2 (Fig. 5a and Supplementary Fig. 11a), involving the loop harboring the catalytic cystine C926 (catalytic loop: residues S924-V938) that accesses the minor groove for stabilizing the flipped Z7, a loop in the RFTS domain (RFTS loop: residues Y199-K220) that contacts the minor groove of the DNA segment downstream of Z7, a loop bridging the catalytic core with the TRD (bridging loop: Y1038-F1059) that cradles the minor groove of the DNA segment upstream of Z7, a loop of the TRD (TRD loop: S1140-L1180) that accesses the major groove, and residues S1097-Y1102 at the N-terminus of the R1104-residing helix in TRD (TRD helix) that interacts with HP1, lining the DNA backbone along the non-target strand (Fig. 5a).

The catalytic loop migrates into the DNA minor groove to interact with both strands of the DNA (Fig. 5a, b). In addition to the interaction with Z7 by DIM2 residues P925, C926, R1013 and R1015 (Fig. 4e), the side chain of DIM2 L932 occupies the space vacated by base flipping of Z7, engaging in van der Waals contacts with the two neighboring bases (Thy6 and Cyt8) (Fig. 5b). Furthermore, DIM2 S930 forms a hydrogen bond with the backbone phosphate of Z7, while DIM2 Q934 donates a hydrogen bond to the sugar ring of orphan guanine Gua7' that otherwise pairs with Z7. In addition, DIM2 Q941 from the α-helix C-terminal to the catalytic loop (denoted as catalytic helix herein) donates a hydrogen bond to the backbone phosphate of Thy9, the side chain of DIM2 L931 makes van der Waals contacts with the base rings of Thy6 and Gua5' in the minor groove, and residue R406 from the RFTS-

BAH1 linker forms a hydrogen-bonding interaction with Cyt10 (Fig. 5b), thereby reinforcing the minor-groove interaction.

Upstream of the target site Z7, the bridging loop of DIM2 runs across both DNA strands, involving residues Y1038, R1039 and K1041 for hydrogen-bonding and/or electrostatic interactions with the backbone phosphate of Cyt5, residues R1043 and N1044 for electrostatic and/or hydrogen-bonding interactions with the backbone of Gua2', and residue N1042 for van der Waals contacts with the backbone of Thy1' (Fig. 5c). Downstream of the target site Z7, residue Y201 from the RFTS loop approaches the minor groove for van der Waals contacts with the Gua11'-Gua10' step, residues Y199 and A200 engage in hydrogen-bonding interactions with the backbone phosphate of Ade16 and Thy15, respectively (Fig. 5d). In addition, the RFTS loop engages in van der Waals contacts with the minor groove, involving DIM2 Y199, Y201, H202, V216, L217, D219, and K220 and both DNA strands (Gua13'-Gua11' and Cyt14-Ade16) (Fig. 5d).

On the major groove side, TRD loop engages in extensive contacts with the nucleotides surrounding orphan guanine Gua7'. Of note, the guanidinium group of DIM2 R1145 forms hydrogen bonds with the O6 atom of Gua7', the N7 atom of Gua8', and the N7 atom of Ade9', while the sidechain carboxylate of DIM2 D1173 forms a bifurcated hydrogen bond with the N1 and N2 atoms of Gua7', and another hydrogen bond with the N4 atom of Cyt8 (Fig. 5e). Furthermore, the guanidium group of DIM2 R1175 intercalates into the non-target strand, prying open the Gua7'-Ade6' base step, as well as forming a salt bridge with the backbone phosphate of Gua7' (Fig. 5e and Supplementary Fig. 11b). Such an R1175-mediated DNA intercalation increased the helical rise by 3.9 Å (7.1 Å for Ade6'-Gua7' vs. 3.2 Å for B-form DNA in Supplementary Fig. 11c), while introduced a roll of −24°, reminiscent of what was previously observed for plant de novo DNA methyltransferase DRM2, in which residue R595 intercalates between the orphan guanine (G$_0$') and the +1-flanking nucleotide (A$_{+1}$') on the non-target strand, leading to substantial DNA deformation (Supplementary Fig. 11d)[58]. In addition, DIM2 T1164 and T1167 engage in hydrogen-bonding interactions with the backbone of Thy6 and Cyt8, respectively, DIM2 K1143 engages in electrostatic contacts with the backbone phosphates of Gua8' and Ade9', and DIM2 T1144 is involved in van der Waals contacts with the backbone of Gua7' (Fig. 5e).

Additional DNA-contact sites of the MTase domain include residues S1097-Y1102 at the N-terminus of the TRD helix, which interact with the non-target strand via hydrogen-bonding and van der Waals contacts, and residues R1208-T1210 at the C-terminus of the MTase domain, which interact with the DNA backbone via salt-bridge and van der Waals interactions (Fig. 5a, e).

To test the observed DIM2-DNA interactions, we mutated key DNA-interacting residues on the RFTS, catalytic loop, bridging loop, and TRD of DIM2 and carried out in vitro DNA methylation analysis. Introducing Y201A mutation to the RFTS domain reduced the DNA methylation efficiency of DIM2 by ~3-fold, introducing the R1039A and R1043A mutations to the bridging loop reduced the DNA methylation efficiency of DIM2 by ~3- and ~8-folds respectively, and introducing the Y1102A, R1145A, and D1173A mutations to the TRD reduced the DNA methylation activity of DIM2 by ~6-, ~8- and ~10-folds, respectively (Fig. 5f). More strikingly, introducing the mutation to the catalytic loop (S930A) or its subsequent helix (Q941A), and T1100A, T1164A, T1166A/T1167A and R1175A mutations to the TRD each led to abolished DNA methylation activity of DIM2 (Fig. 5f). Together, these data lend a strong support for the observed DIM2-DNA interaction.

## Structural basis for RFTS-H3K9me3 and BAH1-H3K9me3 interactions

The RFTS-H3K9me3 interaction is mediated by residues T6-R17 of histone H3K9me3 peptide, which runs through a surface groove formed by the N-terminal and C-terminal lobes of the RFTS domain (Fig. 4d, g). Of note, the side chain of residue H3K9me3 inserts into the

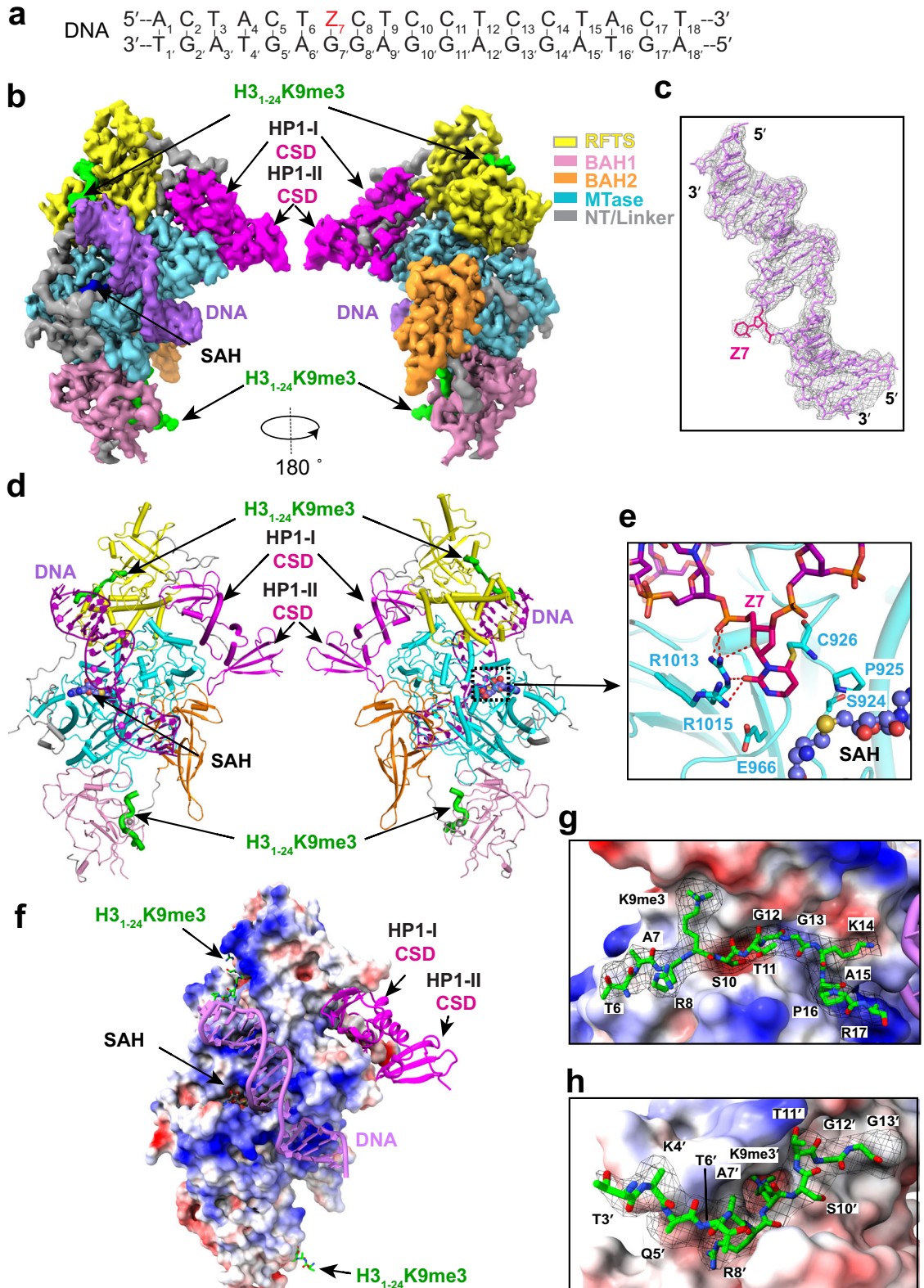

**Fig. 4 | Cryo-EM structure of the DIM2-HP1-H3K9me3-DNA complex. a** DNA sequence used for structural study. **b** Cryo-EM density map for the DIM2-HP1-H3K9me3-DNA complex in two opposite views. The individual domains of DIM2 are color coded. HP1, H3K9me3, and DNA are colored in magenta, lime green, and purple, respectively. **c** Atomic model of the DIM2-bound DNA and the corresponding density map (gray). The flipped-out zebularine Z7 is colored hot pink. **d** Atomic model of the DIM2-HP1-H3K9me3-DNA complex, with individual

components colored in the same fashion as in (**b**). The SAH molecule and zinc ion are shown in sphere representation. **e** Close-up view of the flipped-out Z7 bound to DIM2 residues. The hydrogen bonds are depicted as red dashed lines. **f** Electrostatic surface view of DIM2 bound to HP1, H3K9me3 and DNA. Close-up view of H3K9me3 peptide bound to DIM2 RFTS (**g**) or BAH1 domain (**h**), with the atomic model of the histone peptide covered in the corresponding density map in black mesh and DIM2 shown in electrostatic surface representation.

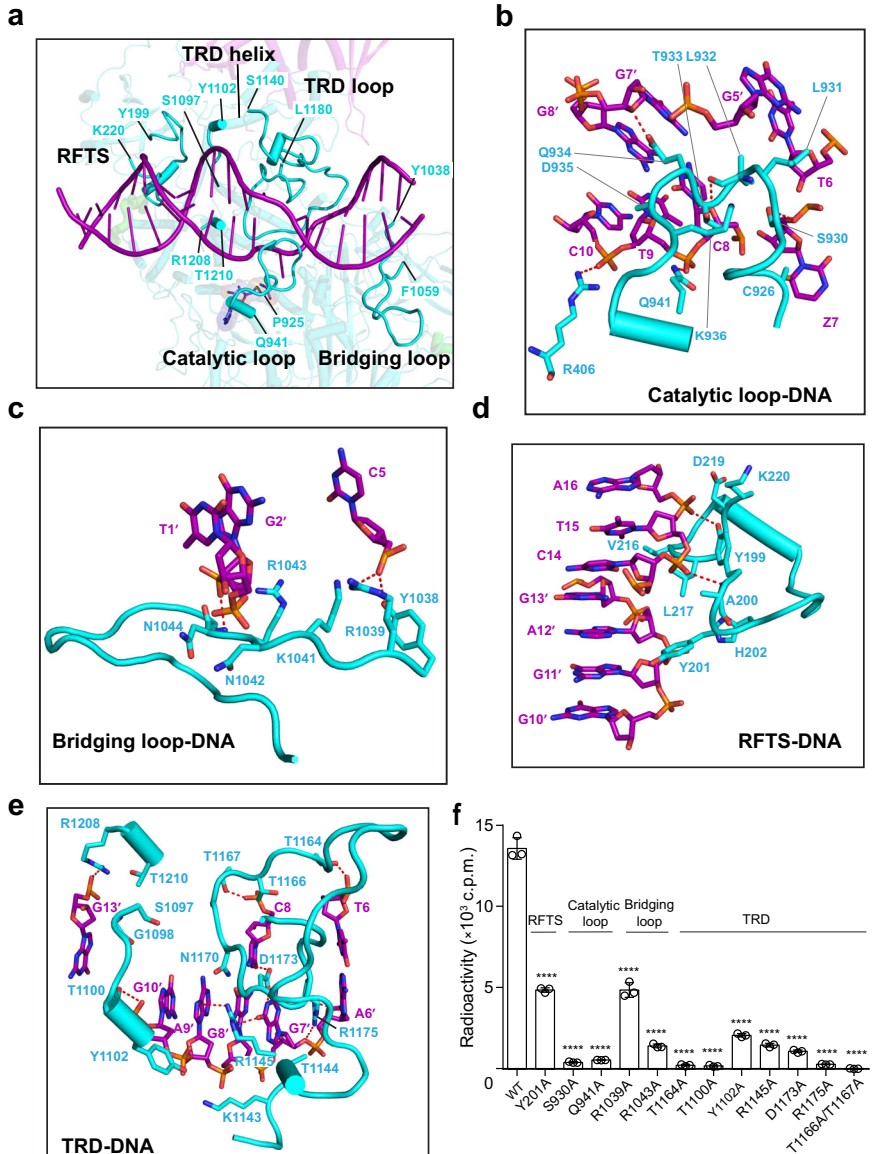

**Fig. 5 | Structural basis of the DIM2-DNA interaction. a** The overall view of intermolecular interactions between DIM2 (cyan) and DNA (purple), with individual DNA-binding regions of DIM2 delimited by residue numbers. Close-up view of the DNA interactions by the catalytic loop (**b**), bridging loop (**c**), RFTS (**d**), and TRD (**e**). The hydrogen bonds are depicted as red dashed lines. **f** In vitro DNA methylation assay of DIM2, WT or DNA binding-associated mutant, on a (CTA)$_{12}$/(TAG)$_{12}$ DNA duplex. The corresponding regions of DIM2 are indicated on top. Data are mean ± s.d. ($n$ = 3 biological replicates). Statistical analysis used two-tailed Student's $t$ test. ****$p < 0.0001$. Source data are provided as a Source Data file.

aromatic cage at the N-terminal lobe, engaging hydrophobic contacts with DIM2 Y185, W261 and Y273 and an electrostatic contact with DIM2 E271 (Fig. 6a). The RFTS-H3K9me3 association is further supported by an array of backbone hydrogen-bonding interactions, involving DIM2 Y249 and H3 A7 and R8, DIM2 N248 and H3 R8, DIM2 V246 and H3 S10, DIM2 L244 and H3 G12, and a side chain-main chain hydrogen bond between DIM2 E363 and H3 S10 (Fig. 6a). In addition, the RFTS-H3K9me3 association is supported by van der Waals contacts involving DIM2 V246, N248, Y249, W361, K362, and H3 T6-R17. Such an RFTS-H3K9me3 association in turn positions H3 K14 in proximity with DNA backbone of Cyt17 for a salt-bridge interaction (Fig. 6a). The fact that the H3K9me3 peptide concurrently interacts with the DIM2 RFTS domain and the DNA substrate explains why the RFTS-H3K9me3 interaction promotes DIM2-mediated DNA methylation.

In addition to the RFTS-H3K9me3 interaction, a second H3K9me3 peptide is anchored to the surface groove of the BAH1 domain, lined by the allosteric loop on one side and the loop connecting β3- and β4-strands (l$_{34}$) of the BAH1 domain on the other side (Fig. 6b). The side chain of H3K9me3 inserts into the hydrophobic cage formed by DIM2 W571, W581, W609, and Y611 for hydrophobic and cation-π interactions, while engaging in an electrostatic interaction with neighboring D615 (Fig. 6b). Furthermore, the DIM2 BAH1 domain interacts with the H3K9me3 peptide via hydrogen-bonding interactions involving DIM2 H634 and H3 S10', DIM2 T559 and H3 R8', and DIM2 N566 and H3 Q5', an electrostatic interaction between DIM2 E649 and H3 R8' (Fig. 6b). In addition, a hydrogen-bonding interaction is formed between DIM2 N1050 of the bridging loop and H3 T11' (Fig. 6b). Structural comparison of the DIM2-HP1-H3K9me3-DNA quaternary complex with the DIM2-HP1 binary complex indicates that in the absence of the H3K9me3 peptide, the bridging loop in the DIM2-HP1 complex would be positioned in steric clash with the DNA substrate (Supplementary Fig. 10c); the H3K9me3 binding induces the structural ordering of the allosteric loop, which in turn moves the bridging loop to a position suitable for DNA binding (Supplementary Fig. 10b, c).

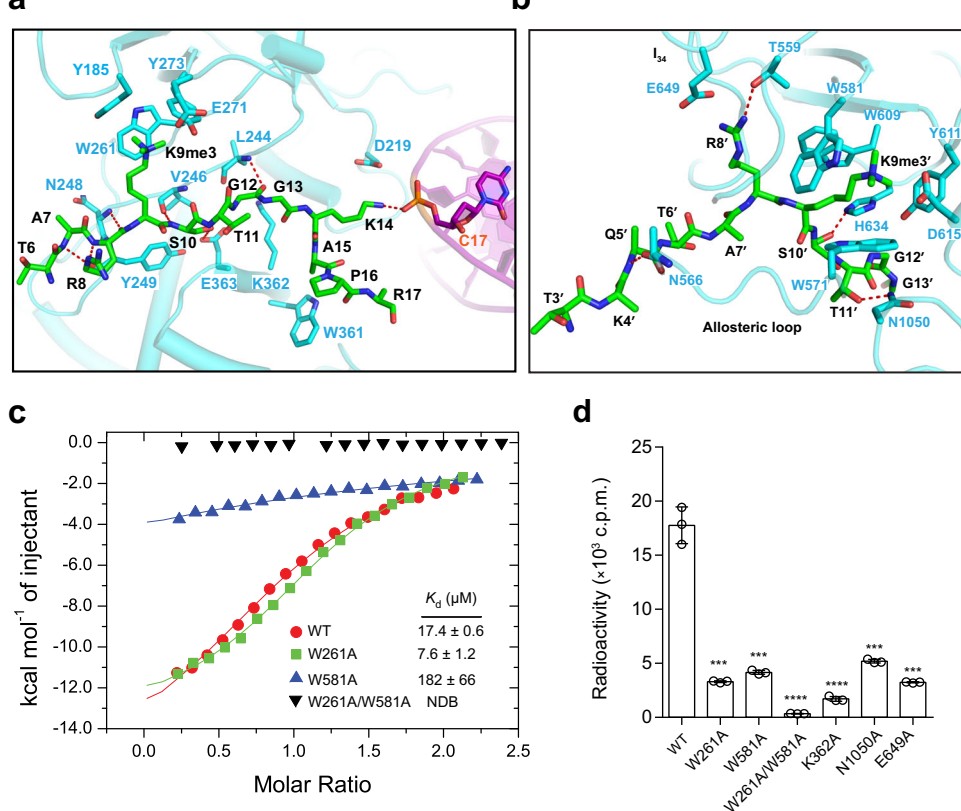

**Fig. 6 | Structural basis for H3K9me3-mediated DIM2 activation. a** Close-up view of the interaction of the H3$_{1-24}$K9me3 peptide with the DIM2 RFTS domain and DNA. The hydrogen bonds are depicted as dashed lines. **b** Close-up view of the interaction between the H3$_{1-24}$K9me3 peptide and the DIM2 BAH1 domain. **c** ITC binding curves for DIM2, WT or mutant, with the H3$_{1-24}$K9me3 peptide. The average and error estimates of the dissociation constants ($K_d$) were derived from two

independent measurements. NDB no detectable binding. **d** In vitro DNA methylation assay of DIM2, WT or H3$_{1-24}$K9me3 binding-associated mutant, on a (CTA)$_{12}$/(TAG)$_{12}$ DNA duplex. Data are mean ± s.d. ($n$ = 3 biological replicates). Statistical analysis used two-tailed Student's $t$ test. ***$p$ = 0.0003; ****$p$ < 0.0001. Source data are provided as a Source Data file.

## Cooperative regulation of DIM2 by the RFTS and BAH1 domains

The observations for (1) the interaction between DIM2 RFTS-bound H3K9me3 peptide and DNA and (2) the H3K9me3 binding-induced repositioning of the bridging loop of the DIM2 MTase domain provide an explanation for the H3K9me3-mediated enzymatic stimulation of DIM2. To test the notion, we mutated the aromatic residues constituting the H3K9me3 cage in the RFTS domain (W261) or BAH1 domain (W581) into alanine and performed ITC binding assays. First, fitting of the ITC binding curve for WT DIM2 with H3K9me3 peptide using one-site binding mode gave an apparent dissociation constant ($K_d$) of 17.4 μM (Fig. 6c and Supplementary Fig. 12a). Next, introducing the W581A mutation to the BAH1 domain reduced the peptide-binding affinity of DIM2 by ~10-fold ($K_d$ of 182 μM), to a level that is comparable with the binding affinity of the isolated DIM2 RFTS domain for the H3K9me3 peptide (Supplementary Fig. 12b, c), whereas introducing the W261A mutation to the RFTS domain led to a -2.3-fold enhanced binding affinity ($K_d$ of 7.6 μM) (Fig. 6c and Supplementary Fig. 12d). Finally, introducing the W261A/W581A double mutation to DIM2 led to complete abolishment of the DIM2-H3K9me3 interaction (Fig. 6c and Supplementary Fig. 12e). These data support the H3K9me3 bindings by the RFTS and BAH1 domains, and suggest that in the absence of DNA, the BAH1-H3K9me3 binding is -24-fold stronger than the RFTS-H3K9me3 binding. However, in the presence of DNA, the RFTS-H3K9me3 binding may be strengthened by the additional interaction between H3 K14 and DNA (Fig. 6a). Presumably owing to such concurrent yet distinct H3K9me3 bindings by the RFTS and BAH1 domains, we found it intractable to fit the ITC binding curve measured for WT

DIM2 using two-site binding mode. In support of the role of the RFTS and BAH1 domains in the DIM2-DNA interaction, our electrophoretic mobility shift assay (EMSA) revealed that both the W261A and W581A mutation modestly reduced the DNA-binding affinity of DIM2 (Supplementary Fig. 12f).

Consistent with the peptide and DNA binding assays, our in vitro DNA methylation analysis reveals that introducing the W261A and W581A mutation each led to 4–5-fold decrease in DNA methylation efficiency of DIM2, whereas introducing the W261A/W581A double mutation largely abolished the DNA methylation activity of DIM2 (Fig. 6d). In addition, mutation of some other H3K9me3-binding sites in the RFTS domain (K362A), the BAH1 domain (E649A) or the bridging loop (N1050A) also led to marked decrease of the DNA methylation activity of DIM2 (Fig. 6d). Together, these data support the notion that separate H3K9me3 bindings by the DIM2 RFTS and BAH1 domains cooperate in allosteric stimulation of DIM2-mediated DNA methylation.

## Structural comparison of DIM2-DNA and DNMT1-DNA complexes

Although DIM2 and DNMT1 share a similar domain architecture, they possess distinct enzymatic specificities, with DIM2 methylating both CG and non-CG DNA (Supplementary Fig. 1f)[23] and DNMT1 strictly methylating CG sites[13,41]. To illustrate how such two evolutionarily linked DNA methyltransferases functionally diverge, we performed structural comparison of DIM2 and DNMT1 (Fig. 7a). DIM2 and DNMT1 shares a similar core structure, formed by the BAH1, BAH2 and

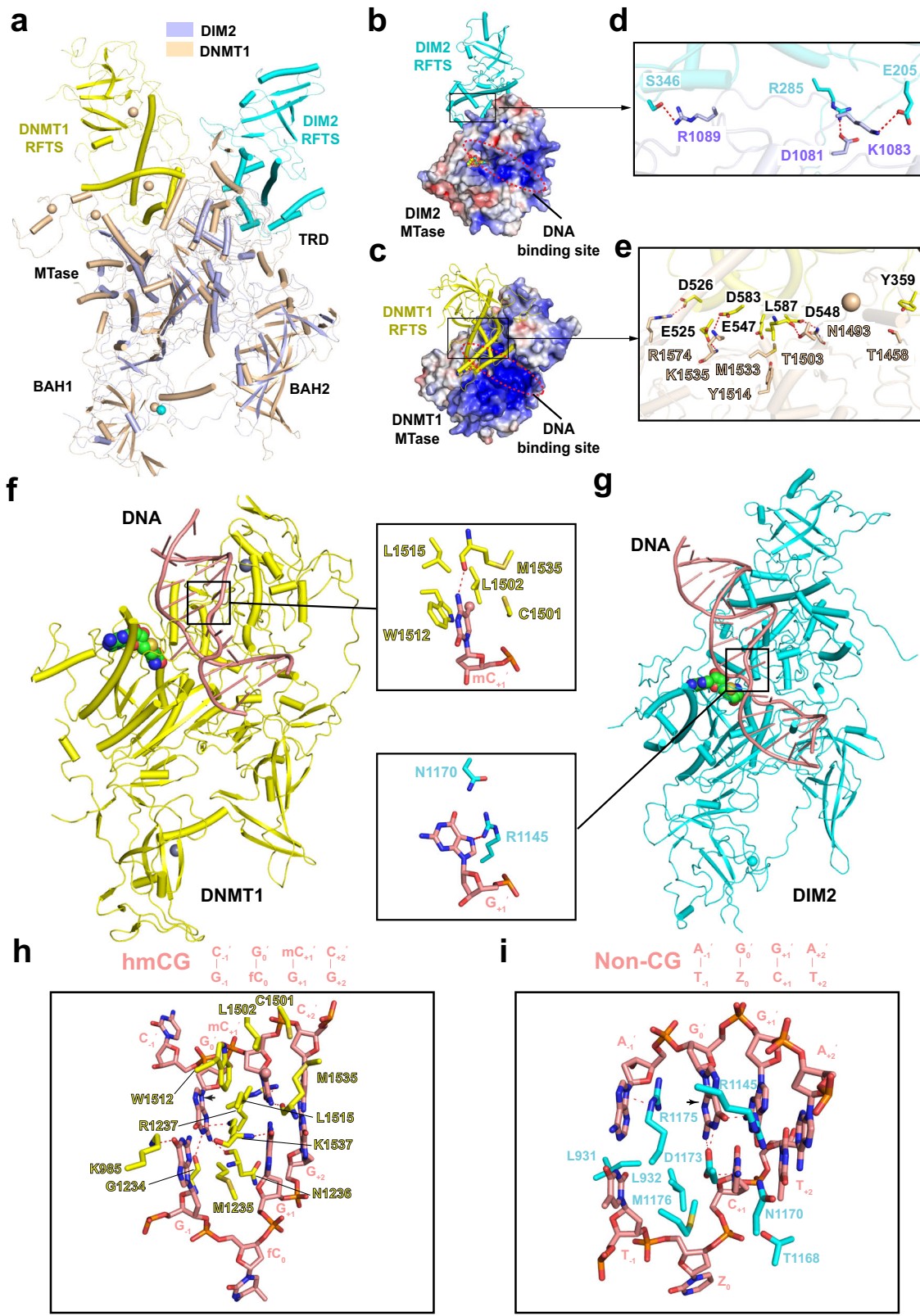

MTase domains, with an RMSD of 4.4 Å over 497 aligned Cα atoms (Fig. 7a), in line with the modest sequence identity between the two orthologues (Supplementary Fig. 13).

The most notable difference between DIM2 and DNMT1 lies in the RFTS domain, which shows drastically different domain conformations: the DIM2 RFTS domain associates with the sidee of the MTase domain to support the DNA binding, whereas the DNMT1 RFTS domain occupies the front face of MTase domain to block DNA binding in DNMT1 (Fig. 7a–c), underpinned the distinct RFTS-MTase interdomain interactions between the two proteins (Fig. 7d, e). Accordingly, whereas the DIM2 and DNMT1 RFTS domains share a similar two-lobe fold, with an RMSD of 4.3 Å over 160 aligned Cα atoms (Supplementary Fig. 14a–c), they possess distinct surface electrostatic potential for inter-domain interactions (Supplementary Fig. 14d, e). Nevertheless,

**Fig. 7 | Structural comparison of DIM2 and DNMT1. a** Structural overlay between the apo forms of DIM2 and DNMT1 (residues 351-1600) (PDB 4WXX). The RFTS domains of DIM2 and DNMT1 are colored in cyan and yellow, respectively. The rest of DIM2 and DNMT1 are colored in slate and wheat, respectively. Positioning of the DIM2 (**b**) or DNMT1 (**c**) RFTS domain (ribbon representation) on top of the MTase domain (electrostatic surface representation). The DNA-binding site of the MTase domain of each protein is marked by a dotted circle. Close-up view of the RFTS-MTase interactions within DIM2 (**d**) or DNMT1 (**e**). The hydrogen bonds are depicted as red dashed lines. **f** Atomic model of DNMT1 (yellow)-DNA (salmon) complex (PDB 4DA4), with the residues surrounding the 5-methylcytosine (mC$_{+1}$′) on the template strand shown in the expanded view. The 5-methyl group of mC$_{+1}$′, zinc ions and SAH molecule are shown in sphere representation. Hydrogen bond is depicted as a dashed line. **g** Atomic model of DIM2 (cyan)-DNA (salmon) complex, with the residues surrounding the +1-flanking nucleotide (G$_{+1}$′) on the complimentary strand shown in the expanded view. Close-up view of the DNA interactions by DNMT1 (**h**) or DIM2 (**i**) MTase domains, centered on at the target site, as well as the −1 to +2-flanking nucleotides. The black arrows indicate the positional shift of Orphan G$_0$′ from an ideal B-form DNA. The hydrogen bonding interactions are shown as dashed lines.

the DIM2 and DNMT1 RFTS domains interact with the H3K9me3 peptide in a similar mechanism: both of the RFTS domains bind to the H3K9me3 peptide via the cleft between the N- and C-lobes of RFTS, lining the H3K9me3 mark by one (W463 in bovine DNMT1) or three (Y185, W261, and Y273 in DIM2) aromatic residues, reinforced by electrostatic contacts with acidic residues (Supplementary Fig. 14b, c). On the other hand, unlike the DNMT1 RFTS-H3K9me3 interaction that involves residues R2-K23 of H3 as well as H3 ubiquitylation, the DIM2 RFTS domain engages a shorter H3K9me3 peptide (residues T6-R17) (Supplementary Fig. 14a–c) and contains a different sequence for the region corresponding to the H3 ubiquitin-binding sites of DNMT1 RFTS (Supplementary Fig. 13).

Structural comparison of DNMT1 and DIM2 further reveals that, unlike the DIM2 TRD that undergoes an extensive disorder-to-order transition upon HP1 binding, the DNMT1 TRD is well defined even in the DNA-free state of DNMT1 (Supplementary Fig. 14f), providing an explanation for the fact that DNMT1 alone, but not DIM2 alone, shows an appreciable methyltransferase activity[59] and in line with the fact that DIM2 TRD and DNMT1 TRD are not conserved in both sequence and structure (Fig. 7a and Supplementary Fig. 13). Along the line, DNMT1 and DIM2 employ a different set of residues for substrate binding (Fig. 7f–i). For instance, DNMT1 inserts three bulky residues (W1512, M1535 and K1537) to occupy the DNA cavity after base flipping of the target cytosine, moving the orphan guanine (G$_0$′) toward the −1-flanking nucleotide on the non-target strand (Fig. 7h), which contributes to the flanking sequence-dependent DNA deformation at the −1 CG-flanking site as observed previously[60]. On the other hand, the DIM2-DNA interaction results in DNA deformation in a different fashion: DIM2 R1175 intercalates between G$_0$′ and −1-flanking nucleotide on the non-target stranding, pushing G$_0$′ toward the +1-flanking nucleotide (Fig. 7i), an opposite direction from that in the DNMT1-DNA complex (Fig. 7h). Furthermore, unlike the DNMT1-induced DNA deformation that is compensated by extensive base-specific hydrogen-bonding interactions involving the CG dyad, which gives rise to the strict CG specificity of DNMT1[37,41], the DIM2-mediated DNA deformation is not accompanied by extensive base-specific interactions (Fig. 7h, i). In addition, DNMT1 TRD forms a hydrophobic concave anchoring the 5-methyl group of 5-methylcytosine on the template strand (Fig. 7f, h), underpinning its substrate preference for hemi-methylated CG sites; in contrast, the corresponding region of DIM2 contains polar residues (N1170 and R1145) in contact with the +1-flanking site (G$_{+1}$′) on the non-target strand (Fig. 7g, i), which may relax the substrate selectivity of DIM2 for hemimethylated CG sites[23]. How the DIM2-DNA interaction interplays with substrate specificity of DIM2, if any, awaits further investigation.

It is worth mentioning that the DIM2 RFTS-BAH1 linker contains a pair of α-helices (αA and αB denoted herein) positioned next to the catalytic helix, resembling the corresponding segments in DNMT1 (Supplementary Fig. 14g). Note that previous studies have demonstrated that the corresponding linker segments in DNMT1, through interaction with the catalytic helix, play a critical role in enzyme activation, allowing for a kinked-to-straight conformational adjustment of the catalytic helix of DNMT1 when transiting from an autoinhibitory state to the active state (Supplementary Fig. 14h)[39,40]. In contrast, the catalytic helix of DIM2 remains straight in both the DNA-free and -bound states (Supplementary Fig. 14i). In this regard, structural analysis of the DNMT1 reveals that the kinked conformation of the catalytic helix is in part stabilized by an inter-domain contact between residue E572 of the RFTS domain and residue R1238 of the catalytic helix (Supplementary Fig. 14j); in contrast, no corresponding interaction is observed for the apo-DIM2 due to a DNMT1-distinct positioning of the RFTS domain, providing an explanation that why the catalytic helix of DNMT1, but not DIM2, undergoes a conformation readjustment between the apo and DNA-binding states[36,37,43,60,61].

Another structural distinction between DIM2 and DNMT1 lies in the BAH1 domain (Supplementary Fig. 15a–c). Structural analysis of the DIM2 BAH1-H3K9me3 association reveals a binding mode similar to that of the previously reported interaction between the BAH domain of maize CHG DNA methyltransferase ZMET2 and the H3K9me2 peptide (Supplementary Fig. 15d, e), rather than the DNMT1 BAH1-H4K20me3 binding (Supplementary Fig. 15b). In fact, structural alignment of the DIM2 BAH1-H3K9me3 and ZMET2 BAH-H3K9me2 complex gives an RMSD of 2.2 Å over 103 aligned Cα atoms (Supplementary Fig. 15e); in contrast, structural alignment of the DIM2 BAH1-H3K9me3 and DNMT1 BAH1-H4K20me3 complex gives an RMSD of 2.4 Å over 79 aligned Cα atoms, with the histone peptides anchored in an opposite polarity (Supplementary Fig. 15c). Together, these data reveal an interplay of DIM2 with DNA and repressive histone marks that is distinct from that of DNMT1.

## Discussion

DNA methylation, H3K9me3, and HP1 protein together serve as the hallmarks of heterochromatin. How they crosstalk to underpin the heterochromatic assembly and genomic stability remains a fundamental question. This study characterized the structure and regulation of DIM2, the sole DNA methyltransferase in the vegetative tissue of *Neurospora* that specifically mediates DNA methylation at constitutive heterochromatin[25]. Importantly, through structure delineation of apo-DIM2, DIM2-HP1 complex, and DIM2-HP1-H3K9me3-DNA complex, combined with biochemical and enzymatic analysis, our work provides an unprecedented view on the functional interplay between heterochromatin factors, in which multivalent readouts of H3K9me3 and HP1 act as a switch in controlling the DNA methylation activity of *Neurospora* DIM2, with important implication in understanding heterochromatin-directed DNA methylation establishment and maintenance.

First, this study reveals a mechanism by which DIM2-mediated DNA methylation is strictly controlled by heterochromatin factors. Unlike most other DNA methyltransferases characterized so far that, in the absence of regulatory proteins, exhibit a basal DNA methylation activity, DIM2 alone possesses no appreciable DNA methylation activity; rather, it requires the presence of both H3K9me3 and HP1 proteins for a substantial DNA methylation activity. Such a tight control of DIM2-mediated DNA methylation is underpinned by three parallel intermolecular interactions: DIM2-HP1, DIM2 RFTS-H3K9me3, and DIM2 BAH1-H3K9me3. First, the DIM2-HP1 interaction leads to two consequences that both contribute to DIM2-mediated DNA methylation in heterochromatin: on one hand, the interaction of HP1 with both

the I130-V132-L134 motif-residing NT segment and the TRD of DIM2 leads to a disorder-to-order transition of the TRD, thereby controlling the TRD-mediated DNA interaction; on the other hand, the simultaneous association of HP1 with DIM2 and H3K9me3 helps recruit DIM2 to H3K9me3-marked heterochromatin, thereby contributing to heterochromatin-specific DNA methylation[62]. Second, the interaction of the DIM2 RFTS domain with H3K9me3 helps position the latter for a DNA interaction, thereby supporting the substrate association of DIM2. Third, the interaction of the DIM2 BAH1 domain with H3K9me3 leads to repositioning of the bridging loop of DIM2, which further facilitates the DIM2-substrate association. These multiple epigenetic regulations provide a mechanism allowing heterochromatin factors H3K9me3 and HP1 to tightly control DIM2-mediated DNA methylation, which in turn reinforces genomic stability at the constitutive heterochromatin.

Second, this study reveals an intricate interplay between HP1 and DNA methylation. As a well-characterized reader for the methylated H3K9 (H3K9me)[4–6], HP1 knowingly employs its homodimeric assembly to promote phase-separated condensates, thereby contributing to chromatin compaction[63–66]. In addition, HP1 proteins reportedly bind to the DNA methyltransferases (DNMTs) across various species, such as plant CMT3[67], and mammalian DNMT1[68,69], DNMT3A[68] and DNMT3B[70]. Whereas these DNMT-HP1 interactions conceivably contribute to the recruitment of respective DNMTs to H3K9me3-enriched heterochromatic regions[71], the molecular basis for the HP1-DNMT axis remains vague. In this regard, this study reveals that the HP1 binding induced an extensive disorder-to-order transition of the DIM2 TRD, thereby coupling the HP1 interaction with DIM2-mediated DNA methylation in heterochromatin. Such an HP1-mediated regulation lends an explanation to a previous observation that disruption of the HP1 interaction of DIM2 led to a loss of DNA methylation in *Neurospora*[32], providing a mechanism for heterochromatin-specific DNA methylation by DIM2. Interestingly, the HP1 binding also appears to stimulate the DNA methylation activity of DNMT1[69]. Whether the HP1-mediated regulation of DIM2 is shared with DNMT1 and other DNMTs awaits further investigation.

Third, this study adds another example on the functional crosstalk between DNA methylation and repressive histone modifications, which represents a conserved epigenetic mechanism in reinforcing heterochromatin assembly and gene silencing[26]. Direct recognition of H3K9me3 and H4K20me3 has been observed for the RFTS and BAH1 domains of DNMT1, respectively, leading to allosteric activation of its DNA methylation activity[29,35]. For CMT3, it employs both the

Chromodomain and BAH domain to interact with H3K9me2[27,28], with the Chromodomain-H3K9me2 binding mediating the chromatin targeting and the BAH-H3K9me2 interaction leading to allosteric activation[28,30]. Our study reveals that both the RFTS and BAH1 domains of DIM2 recognize H3K9me3, which in turn contribute to the association between DIM2 and DNA substrates through direct DNA contact (for RFTS-bound H3K9me3 peptide) or conformational adjustment of a DNA-binding site (for BAH1-bound H3K9me3 peptide). It remains to be investigated that whether the two DIM2-bound H3K9me3 marks arise from a di-nucleosome, as that demonstrated for CMT3[30] (Fig. 8), or a mono-nucleosome. Nevertheless, these observations add an example of the functional crosstalk between DNA methylation and H3K9me3.

Fourth, this study uncovers how closely related DNMT1 and DIM2, with a common scaffold, have evolved into distinct functional specificities. For DNMT1, its RFTS domain interacts with the front face of the MTase domain to inhibit its DNA methylation activity[38–40]; the association of H3 two-monoubiquitylation/H3K9me3 with the RFTS domains promotes the repositioning of the latter, thereby relieving the autoinhibitory state[29,42,43]. In contrast, the DIM2 RFTS domain is positioned on the side face of the MTase domain, allowing the DNA binding interface of the latter fully accessible to DNA substrates. The association of H3K9me3 with the RFTS domain did not lead to domain repositioning of the latter, rather, it anchors the H3K9me3 peptide for a direct interaction with the DNA substrates, thereby supporting the DIM2-DNA association. Furthermore, unlike the DNMT1 BAH1 domain that specifically recognizes H4K20me3, this study reveals that the DIM2 BAH1 domain interacts with H3K9me3 peptide in a manner resembling that of ZMET2 BAH domain-H3K9me2 interaction. The DIM2 BAH1-H3K9me3 interaction allosterically activates DIM2 via conformational adjustment of a DNA-binding loop of the MTase domain, which is again reminiscent of ZMET2 BAH-H3K9me2 interaction[28].

In addition, this study reveals that DIM2 interacts with the DNA substrates in a manner different from that of the DNMT1-DNA interaction. The DNMT1-DNA interaction involves a network of base-specific hydrogen-bonding interactions involving the CG site, as well as a hydrophobic concave for anchoring the 5-methyl group of 5-methylcytosine on the hemimethylated CG site[37,60]. These base-specific contact sites are not present in DIM2, due to lack of sequence similarity between the TRDs of DIM2 and DNMT1. On the other hand, the DIM2-DNA interaction involves strong DNA intercalation induced by an arginine residue (R1175), resulting in base unstacking between

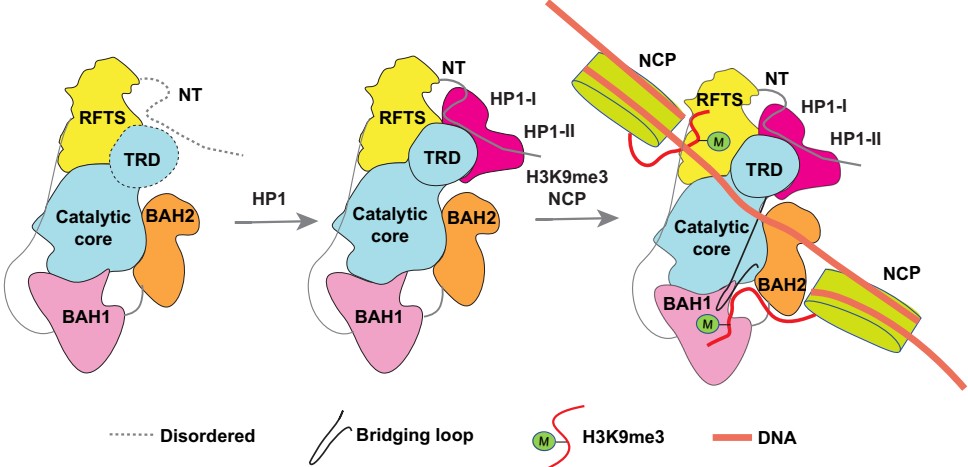

**Fig. 8 | Multi-layered regulation of DIM2-mediated DNA methylation.** A working model for the multi-layered regulation of DNA methylation by DIM2. The binding of HP1 induces a disorder-to-order transition of the DIM2 TRD. The dual readout of histone H3K9me3 by the DIM2 RFTS and BAH1 domains allosterically activates DIM2-mediated DNA methylation within the nucleosome environment. NCP nucleosome core particle. It remains to be investigated whether the two H3K9me3 marks arise from two adjacent nucleosomes or from the same nucleosome.

the orphan guanine and the −1-flanking nucleotide. A similar intercalation-induced DNA deformation was previously observed for plant DNA methyltransferase DRM2, which gives rise to a substrate preference of DRM2 for a CHH site containing an A/T nucleotide at the +1 position[58]. Whether the DIM2 R1175-induced DNA deformation confers DIM2, which also targets genomic regions enriched with A/T nucleotides, a flanking sequence preference awaits further investigation. Together, these observations provide insight into the evolutionary conservation and divergence among DNMT1 family of DNA methyltransferases.

In summary, this study provides insight into an intricate interplay between heterochromatic factors in regulating DNA methylation, and how DIM2, with a DNMT1-like scaffold, has evolved to orchestrate the unique DNA methylation landscape in *Neurospora*.

## Methods

### Protein expression and purification

DNA encoding *Neurospora Crassa* DIM2 (NCBI accession no. XP_959891.1) (residues 1-1242) was derived from a genomic DNA library of *Neurospora Crassa* by PCR amplification. The DNA fragment encoding *Neurospora Crassa* HP1 (residues 1-266; uniport number. Q1K612) was chemically synthesized by Integrated DNA Technologies, codon optimized for bacterial expression. Both the DNA fragments for DIM2 and HP1 were cloned into an in-house vector preceded by an N-terminal His$_6$-MBP tag and a TEV cleavage site. The expression plasmids were transformed into BL21 (DE3) RIL cells strain (Agilent Technologies). The transformed cells were first grown at 37 °C until OD$_{600}$ reached 1.0. Afterwards, the temperature was shifted to 16 °C and the cells were induced by addition of 0.13 mM isopropyl β-D-1-thiogalactopyranoside (IPTG) and continued to grow overnight. The cells were harvested and lysed in buffer containing 50 mM Tris-HCl (pH 8.0), 1 M NaCl, and 25 mM imidazole. After centrifugation, the His$_6$-MBP-DIM2 or His$_6$-MBP-HP1 fusion protein was purified from the supernatant through a Ni-NTA column (Cytiva), followed by ion-exchange chromatography on a HiTrap Heparin HP column (Cytiva), removal of the His$_6$-MBP tag by TEV-mediated proteolytic cleavage, and a second round of Ni-NTA chromatography. The tag-free proteins were further purified by size exclusion chromatography on a Superdex 200 16/600 column (GE Healthcare) pre-equilibrated with a buffer containing 20 mM Tris-HCl (pH 7.5), 100 mM NaCl, 5% glycerol and 5 mM DTT. Purified protein samples were stored at −80 °C in a concentration of ~6 mg/mL for future use.

For biochemical analysis of the DIM2 RFTS domain, the DNA encoding residues 1-389 of DIM2 was inserted into the expression vector described above, and the protein was purified in the same manner as that for residues 1-1242 of DIM2, except that a Superdex 75 16/600 column (GE Healthcare) was used for final protein purification. The DIM2 mutants were generated by site-directed mutagenesis and purified using the same approach as described for WT protein.

### In vitro DNA methylation assay

In vitro DNA methylation assay was performed in a 20-μL reaction mixture. For the substrate specificity assay, the following DNA duplexes were used as substrates: CG DNA (Upper strand: 5′-ATTAT-TAATCGAAATTTA-3′), hemimethylated CG DNA (Upper strand: 5′-TAAATTTXGATTAATAAT-3′, X = 5-methylcytosine; Lower strand: 5′-ATTATTAATCGAAATTTA -3′), CHG DNA (Upper strand: 5′-ATTAT-TAATCTGAATTTA-3′), and CHH DNA (Upper strand: 5′-ATTAT-TAATCTAAATTTA-3′). The reaction mixture contains 0.25 μM DIM2, 1.0 μM DNA duplex, 1.0 μM synthesized H3K9me3 peptide (residues 1-24 of H3 plus a C-terminal tryptophan), 0.5 μM HP1, and 0.56 μM S-adenosyl-L-[methyl-$^3$H] methionine with a specific activity of 18 Ci/mmol (PerkinElmer) in buffer containing 50 mM Tris-HCl (pH 8.0),

100 mM NaCl, 0.05% β-mercaptoethanol, 5% glycerol, and 200 μg/mL BSA. For the rest of the DNA methylation assay, each reaction mixture contains 1.0 μM DIM2, 1.0 μM (CAT)$_{12}$/(ATG)$_{12}$ DNA duplex and 0.56 μM S-adenosyl-L-[methyl-$^3$H] methionine with a specific activity of 18 Ci/mmol (PerkinElmer) in buffer containing 50 mM Tris-HCl (pH 8.0), 100 mM NaCl, 0.05% β-mercaptoethanol, 5% glycerol, and 200 μg/mL BSA. In addition, 1.0 μM synthesized H3K9me3 or H3K9me0 peptide (residues 1-24 of H3 plus a C-terminal tryptophan) and/or 2 μM HP1 protein was included to evaluate the effect of H3 and HP1 regulation. The reaction mixture was incubated at 30 °C for 30 min, unless indicated otherwise, before being quenched by the addition of 5 μL of 10 mM cold SAM. After the reaction, 8 μL of the mixtures was loaded onto Hybond N nylon membrane (GE Healthcare), which was left to dry out at room temperature. The membrane was subsequently washed with 0.2 M ammonium bicarbonate (pH 8.2) two times (20 min per time), deionized water (20 min) once, and 95% ethanol (5 min) once. After air dried, the membrane carrying each sample was transferred into a vial containing 3 mL scintillation buffer (Fisher). The tritium scintillation was measured and recorded by a Beckman LS6500 counter. Each the reaction was repeated three times.

### Analytical size-exclusion chromatography

For size-exclusion chromatography analysis, 400 μL of the sample mixture containing 1.0 μM DIM2 and/or 2.5 μM HP1 was loaded onto Superdex 200 increase 10/300 GL column (GE Healthcare) and eluted at a flow rate of 0.5 mL/min, in the buffer containing 20 mM Tris-HCl (pH 7.5), 100 mM NaCl, 5% glycerol, and 5 mM DTT. The proteins in each fraction were then visualized by SDS-PAGE.

### ITC binding assay

ITC measurements were performed using a MicroCal iTC200 instrument (GE Healthcare). To measure the bindings between proteins and H3K9me3 peptide, 0.04−0.05 mM DIM2, WT or mutant (W261A, W581A, or W261A/W581A), DIM2 RFTS domain, or HP1, WT or mutant (W98A), was titrated with 0.5 mM histone peptide at 7 °C. Prior to the titration, both peptide and protein samples were subjected to overnight dialysis against buffer containing 20 mM Tris-HCl (pH 7.5), 100 mM NaCl, and 1 mM β-mercaptoethanol. Buffer to buffer titration was performed to ensure no abnormality of baseline. Analyses of all data were performed with MicroCal Origin software, fitted with single-site binding mode.

### Thermal shift assay

Thermal shift assays were performed for WT and R1104A DIM2 proteins using a Bio-Rad CFX Connect Real-Time PCR Detection System. Each 20-μL reaction mixture contained 5 μM WT or R1104A-mutated DIM2, and/or 10 μM HP1 protein, dissolved in buffer containing 20 mM Tris-HCl (pH 7.5), 100 mM NaCl, 5% glycerol, 5 mM DTT, and 1× GloMelt Dye (Biotium). The sample plates were subject to heat from 4 to 95 °C with a stepwise increment of 5 °C. Fluorescence intensity was recorded with the excitation and emission wavelength of 470 nm and 510 nm, respectively. The experiments were carried out in triplicate.

### Electrophoretic mobility shift assay

Each 10-μL reaction mixture contained 0.1 μM DNA duplex (Upper strand: 5′-ATTATTAATCTAAATTTA-3′) mixed with 0, 0.2, or 0.4 μM DIM2, WT or mutant (W261A or W581A), and/or 0.8 μM H3K9me3 peptide in binding buffer containing 20 mM Tris-HCl (pH 8.0), 75 mM NaCl, 10% glycerol, 5 mM DTT, and 2% NP-40. Samples were resolved on a 5% w/v polyacrylamide gel (59:1 for acrylamide:bis-acrylamide) with the running buffer containing 1 × Tris-Glycine (pH 8.6) at 4 °C for 1 h. The gel was stained with SYBR™ Gold (Thermo Fisher Scientific) for 10 min and visualized by ChemiDoc Imaging System (Bio-Rad).

## Cryo-EM data collection

For DIM2-HP1 complex, aliquots of 3 μL of DIM2-HP1 complex after gel filtration in a buffer containing 20 mM Tris-HCl (pH 7.5), 100 mM NaCl, 2.5% Glycerol, and 5 mM DTT were applied to glow-discharged Quantifoil® (1.2/1.3) grids. The grids were blotted for 6 s at 95% humidity with an offset of 5 s and plunge frozen into liquid ethane using a Vitrobot Mark IV (Thermo Fisher). Grids were imaged on a 300 keV Titan Krios cryo-electron microscope (Thermo Fisher) equipped with a K3 camera (Gatan) by Pacific Northwest Center for Cryo-EM (PNCC). Movies were collected at a calibrated magnification of ×81,000, corresponding to a 1.056 Å per physical pixel. The dose was set to a total of 67 electrons/Å$^2$ over an exposure of 68 frames. Automated data collection was carried out using SerialEM with a nominal defocus range set from −0.8 to −2.5 μm. A total of 4009 movies were collected over 24 h.

For apo-DIM2, aliquots of 3 μL of DIM2 in a dialysis buffer containing 20 mM Tris-HCl (pH 7.5), 100 mM NaCl, 2.5% Glycerol, and 5 mM DTT were applied to glow-discharged Quantifoil® (1.2/1.3) grids. The grids were blotted for 6 s at 95% humidity and plunge frozen into liquid ethane using a Vitrobot Mark IV (Thermo Fisher). Grids were imaged on a 300 keV Titan Krios cryo-electron microscope (Thermo Fisher) equipped with a K3 camera (Gatan) by NCI cryo-EM Facility (NCEF). Movies were collected at a calibrated magnification of ×81,000, corresponding to a 1.07 Å per physical pixel. The dose was set to a total of 50 electrons/Å$^2$ over an exposure of 40 frames. Automated data collection was carried out using SerialEM with a nominal defocus range set from −0.6 to −1.5 μm. A total of 8327 movies were collected over 48 h.

For DIM2-HP1-H3K9me3-DNA complex, aliquots of 3 μL of DIM2-HP1-H3K9me3-DNA complex after reaction in a buffer containing 50 mM Tris-HCl (pH 8.0), 2.5% Glycerol, and 40 mM DTT were applied to glow-discharged Quantifoil® (1.2/1.3) grids. The grids were blotted for 6 s at 95% humidity and plunge frozen into liquid ethane using a Vitrobot Mark IV (Thermo Fisher). Grids were imaged on a 300 keV Titan Krios cryo-electron microscope (Thermo Fisher) equipped with a K3 camera (Gatan) by Pacific Northwest Center for Cryo-EM (PNCC). Movies were collected at a calibrated magnification of ×81,000, corresponding to a 1.056 Å per physical pixel. The dose was set to a total of 70 electrons/Å$^2$ over an exposure of 30 frames. Automated data collection was carried out using SerialEM with a nominal defocus range set from −0.8 to −2.5 μm. A total of 3728 movies were collected over 24 h.

## Image processing

The cryo-EM data for the DIM2-HP1 complex, apo-DIM2, and the DIM2-HP1-H3K9me3-DNA complex, containing 4009, 8327 and 3728 movies, respectively, were processed using the CryoSPARC software package (v4.01)[72]. All movies were motion-corrected using patch motion correction, and the resulting micrographs were subject to patch-based contrast transfer functions (CTFs) estimation. Subsequently, peak picking was carried out using the Topaz program[73].

For the DIM2-HP1 complex, 1.5 million particles were extracted, followed by several rounds of 2D classifications. One million particles were then selected based on visible features of secondary and tertiary structures. Next, three ab initio models were generated and subsequently heterogeneously refined. The particles associated with one class with well-defined feature of DIM2 and HP1 were selected for non-uniform refinement, resulting in a final resolution of 2.76 Å.

For apo-DIM2, 4.8 million particles were extracted for 2D classifications. Among them, 2.9 million particles, with a down-scaled pixel size of 2.14 Å, were selected after removal of obvious junk and ice. Two rounds of ab initio reconstruction and heterogeneous refinement were carried out, with three models in the first round and six models in the second round. For each round, the particles for the classes with well-

defined secondary and tertiary structures were selected. After the second round, the selected particles were re-extracted with a pixel size of 1.07 Å and used to generate another four ab initio models, followed by another round of heterogeneous refinement. These particles associated with class with the highest resolution were then subject to non-uniform refinement, resulting in a final resolution of 2.88 Å.

For the DIM2-HP1-H3K9me3-DNA complex, 1.6 million particles were extracted with a down-scaled pixel size of 2.11 Å. After 2D classifications, 1.1 million particles were selected based on visible feature of secondary and tertiary features. Five ab initio models were generated and subsequently heterogeneously refined. The particles associated with the class with defined features of DIM2 were re-extracted with pixel size of 1.06 Å. Next, the particles were subject to 3D classification into three models, followed by heterogeneous refinement. The particles associated with the class with defined feature of DIM2 and DNA molecules were selected for further CTF refinement and non-uniform refinement, leading to a final resolution of 2.79 Å. The map was further improved using the anisotropic sharpening module in Phenix (v1.20.1-4487)[74].

## Model building and refinement

The structural models of DIM2 and HP1 were predicted using the AlphaFold2 server[75]. The structural model for the DNA and H3K9me3 peptide were derived from that in the DNMT3A-DNMT3L-DNA complex (PDB 6F57)[56] and that in the DNMT1 RFTS-H3K9me3-ubiquitn complex (PDB 6PZV)[29], respectively. Fitting of the structural models of DIM2, HP1 and/or DNA into the individual maps were performed using UCSF Chimera (v1.16)[76] and ChimeraX (v1.6.1)[77]. The final model was then obtained after iterative model building in Coot (v0.8.9.1)[78] and real-space refinement in Phenix. The RMSD for pairwise protein structural alignment is calculated using the online TM-align method[79] (https://www.rcsb.org/alignment).

## Reporting summary

Further information on research design is available in the Nature Portfolio Reporting Summary linked to this article.

## Data availability

All data needed to evaluate the conclusions in the paper are present in the paper and/or the Supplementary Materials. Coordinates and structure factors for DIM2-HP1, apo-DIM2 and DIM2-HP1-H3K9me3-DNA have been deposited in the Protein Data Bank under accession codes 9BAZ, 9BAP, and 9BAQ, respectively. The cryo-EM density maps have been deposited to the EMDB and PDB under the accession numbers of EMD-44415, EMD-44110, and EMD-44111, respectively. Atomic coordinates used in this study are publicly available from the PDB under accession codes 4WXX, 4DA4, 6F57, and 6PZV. Source data are provided with this paper.

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

## Acknowledgements

This work was supported by NIH (R35GM119721 to J.S.). Cryo-EM data were collected with the assistance of Drs. Rose Marie Haynes and Tara Fox. A portion of this research was supported by NIH grant U24GM129547 and performed at the PNCC at OHSU and accessed through EMSL (grid.436923.9), a DOE Office of Science User Facility sponsored by the Office of Biological and Environmental Research. This research was, in part, supported by the National Cancer Institute's National Cryo-EM Facility at the Frederick National Laboratory for Cancer Research under contract 75N91019D00024. Molecular graphics and analyses performed with UCSF ChimeraX, developed by the Resource for Biocomputing, Visualization, and Informatics at the University of California, San Francisco, with support from National Institutes of Health R01-GM129325 and the Office of Cyber Infrastructure and Computational Biology, National Institute of Allergy and Infectious Diseases. We thank the Central Facility for Advanced Microscopy and Microanalysis of UC Riverside for facility access. We thank Dr. Katherine Borkovich for providing the genomic DNA library of *Neurospora Crassa* for cloning of DIM2.

## Author contributions

Z.S., J.L., and N.K. performed experiments. J.S. conceived the project and supervised the study. Z.S. and J.S. wrote the manuscript with input from all the authors.

## Competing interests

The authors declare no competing interests.
