## [Peer Review File · Nature Communications]

Multi-layered heterochromatin interaction as a switch for DIM2-mediated DNA methylationREVIEWER COMMENTS

Reviewer #1 (Remarks to the Author):

This study uncovered the regulation of the DIM2 DNA methyltransferase by the heterochromatin factors HP1 and H3K9me3. By extensive biochemical and structural studies, the authors showed the structural basis of the activation of DIM2 by HP1 and H3K9me3. To me, one of the most important contributions is that the study suggested the role of H3K9me3 is to directly interact with and regulate DIM2 but not indirectly through HP1. The canonical H3K9me3 reader HP1 here serves as a regulatory protein for DIM2 but does not help DIM2 to H3K9me3 loci. Overall, the manuscript is of good quality, and the study is important to the DNA methylation field. I fully support the publication of the paper. Some minor concerns are listed below.

Although HP1 helps the activation of DIM2 and DIM2 directly binds to H3K9me3, HP1 itself can directly bind to H3K9me3. The role of H3K9me3 binding capacity of HP1 and its potential role in DNA methylation deserve more discussion.

L380, and L387, the RMSDs are 4.2 and 7.8. The structures are similar, but the RMSDs are so high. It is worth to be double-checked. The author also needs to note here how to align the two structures, especially for the later one. The RMSD is calculated by simply overlaying the two RFTSes or by overlaying the core regions but only calculating the RMSD of RFTSes. And also in L440, I cannot believe the RMSD is as high as 16.8 Å, as the two structures look similar.

Reviewer #2 (Remarks to the Author):

This manuscript reports the cryo-EM structures of a DNA methyltransferase DIM2 from *Neurospora* in the absence and presence of bound HP1, histone H3K9me3 peptide, and DNA, including apo-form DIM2, DIM2-HP1 complex, and DIM2-HP1-H3K9me3-DNA complex. This study provides a fundamental basis revealing how DIM2 interacts with HP1 and histone H3K9me3 peptides to methylate DNA at heterochromatin. However, some concerns and suggestions are listed below.

1. The major concern is that the cryo-EM structures cannot explain how DIM2 is strictly regulated by HP1 and H3K9me3. The biochemical assays show that DIM2-mediated DNA methylation is strictly controlled by HP1 and H3K9me3 peptide, and DIM2 requires the presence of both H3K9me3 and HP1 proteins for a substantial DNA methylation activity (Figure 1b). However, the cryo-EM structure of DIM2 is highly similar to those ones in the DIM2-HP1 and DIM2-HP1-H3K9me3-DNA complexes with only minor differences in the TRD loop (disorder to order), RFTS domain (a loop reposition slightly for DNA interaction), and the bridging loop. This minor differences cannot explain the strict control of DIM2 by HP1 and H3K9me3. The authors need to consider other possibilities, such as that the cryo-EM structures of DIM2 alone and DIM2-HP1 complex are likely in the “activated” but not in the “auto-inhibitory” conformation. The previous studies in mammalian DNMT1 demonstrated that RFTS domain in DNMT1 is flexible with two conformations, activated and auto-

inhibitory conformations, and the H3 binding induces the RFTS domain to shift from auto-inhibitory to activated conformation. In other words, the molecular mechanism of the interplay between DIM2, HP1 and H3K9me3 is likely not fully revealed by this study.

2. The DIM2 mutation of R1104A at the TRD loop reduced the methyltransferase activity of DIM2 by 8-fold compared to the wild-type DIM2 (Figure 3h). It is concluded that “This observation therefore supports the notion that HP1 allosterically regulates the DNA methylation activity of DIM2 via structural stabilization of the DIM2 TRD”. It is actually not known if R1104A mutation could reduce or enhance the stability of DIM2 TRD loop. The assertion that HP1 binding enhances the stability of the TRD loop, consequently increasing DIM2 activity, lacks validity.

3. The ITC binding results are strange. Introducing the W261A mutation in the RFTS domain led a ~2.3-fold enhanced binding affinity (Fig. 6c and page 16). This result does not support the cryo-EM structure showing that the methylated K9 in the H3 peptide interacts with W261 in the RFTS domain stated in page 16. Is there other evidence to support that the H3 peptide is bound in the RFTS domain of DIM2?

4. The H3 peptide binding to DIM2 may be affected by the presence of DNA. To clearly elucidate the interplay among DIM2, H3 peptide and DNA, it is suggested to measure the affinity between DIM2 (WT verse mutants) and H3 peptide in the presence of DNA, or measure the DNA-binding affinity of DIM2 (WT verse mutants) in the presence of the H3 peptide.

Minor points :

1. The interactions between DIM2 and HP1 in the Interface I shown in Figure 2c are too busy to reveal any information.

2. Figure 8 is not referred in the main text.

Reviewer #3 (Remarks to the Author):

I co-reviewed this manuscript with one of the reviewers who provided the listed reports. This is part of the Nature Communications initiative to facilitate training in peer review and to provide appropriate recognition for Early Career Researchers who co-review manuscripts

Reviewer #4 (Remarks to the Author):

DNA methylation and histone H3K9me3 play pivotal roles in heterochromatin formation, which is essential for gene regulation and genome stability. However, the molecular mechanisms underlying the crosstalk between DNA methylation and repressive histone modifications are largely unknown. In this study, Shao et al. focused on the structural basis of H3K9me3-binding dependent DNA methylation of DIM2 in *Neurospora*, which belongs to the superfamily of mammalian DNMT1. DIM2

consists of an N-terminal tail (NT), RFTS, BAH1, BAH2, and methyltransferase (MTase) domains. DIM2-dependent DNA methylation and heterochromatin formation are required for HP1 binding, which is an H3K9me3 reader protein. To reveal the molecular basis for DIM2 activation by HP1 and H3K9me3, the authors determined the cryo-EM structures of apo-DIM2, the DIM2-HP1 complex, and the DIM2-HP1-H3K9me3 complex. I am very impressed with the many findings of this study; direct HP1-binding contributes to the structural formation of the TRD of the MTase domain in DIM2. The DNMT1 RFTS domain binds to two-monoubiquitinated histone H3, which is essential for the recruitment of DNMT1 to the methylation site and DNMT1-activation. In contrast, the RFTS and BAH1 domains of DIM2 simultaneously bind to H3K9me3, which is required for the activation of DIM2. This different regulation of DIM2 by H3K9me3 is fundamental to the regulation of DIMA2 activation by the repressive histone mark. Overall, this study clearly revealed H3K9me3 and HP1 binding dependent DIM2 regulation. These results are solid and include the novelty of the regulation of DNA methylation. This paper deserves to be published in Nature Communications. Before publication, the authors are encouraged to improve the messy figures, especially Figures 2c, 5b, 6a, and 6b, to make them clearer and easier to understand.

General Response

We thank all four reviewers for their collective efforts in reviewing our manuscript, their positive view of our work, and their constructive comments for improving the manuscript. As outlined below in the point-by-point response (marked in blue), we have now systematically addressed all the raised critiques and have incorporated them in the revised manuscript (marked in red).

Point-by-point Response

Response to Reviewer 1

Reviewer #1 (Remarks to the Author):

This study uncovered the regulation of the DIM2 DNA methyltransferase by the heterochromatin factors HP1 and H3K9me3. By extensive biochemical and structural studies, the authors showed the structural basis of the activation of DIM2 by HP1 and H3K9me3. To me, one of the most important contributions is that the study suggested the role of H3K9me3 is to directly interact with and regulate DIM2 but not indirectly through HP1. The canonical H3K9me3 reader HP1 here serves as a regulatory protein for DIM2 but does not help DIM2 to H3K9me3 loci. Overall, the manuscript is of good quality, and the study is important to the DNA methylation field. I fully support the publication of the paper. Some minor concerns are listed below.

We thank the reviewer for his/her positive assessment of our work and support for publication of this manuscript. We have addressed the reviewer's minor concerns as below.

Although HP1 helps the activation of DIM2 and DIM2 directly binds to H3K9me3, HP1 itself can directly bind to H3K9me3. The role of H3K9me3 binding capacity of HP1 and its potential role in DNA methylation deserve more discussion.

We thank the reviewer for the suggestion. Indeed, the direct interaction of HP1 with both H3K9me3 and DIM2 not only leads to allosteric activation of DIM2, but also helps recruit DIM2 to H3K9me3-enriched heterochromatin for region-specific DNA methylation. In the revised manuscript, we have included additional discussion (Line 473-476 and Line 477-479) to clarify this point.

L380, and L387, the RMSDs are 4.2 and 7.8. The structures are similar, but the RMSDs are so high. It is worth to be double-checked. The author also needs to note here how to align the two structures, especially for the later one. The RMSD is calculated by simply overlaying the two RFTSes or by overlaying the core regions but only calculating the RMSD of RFTSes. And also in L440, I cannot believe the RMSD is as high as 16.8 Å, as the two structures look similar.

We thank the reviewer for pointing out this issue. We have recalculated the RMSDs for the aligned core using the online TM-align server in the protein data bank (<https://www.rcsb.org/alignment>). The RMSD for the core fragments of DNMT1 and DIM2 is 4.4 Å over 497 aligned C α atoms, the RMSD for the RFTS domains of DIM2 and DNMT1 is 4.3 Å over 160 aligned C α atoms, the RMSD for the BAH1 domains of DIM2 and ZMET2 is 2.2 Å over 103 aligned C α atoms, and the RMSD

for the BAH1 domains of DIM2 and DNMT1 is 2.4 Å over 79 aligned C α atoms. We have clarified this in the text and method section in the revised manuscript.

Response to Reviewer 2

Reviewer #2 (Remarks to the Author):

This manuscript reports the cryo-EM structures of a DNA methyltransferase DIM2 from *Neurospora* in the absence and presence of bound HP1, histone H3K9me3 peptide, and DNA, including apo-form DIM2, DIM2-HP1 complex, and DIM2-HP1-H3K9me3-DNA complex. This study provides a fundamental basis revealing how DIM2 interacts with HP1 and histone H3K9me3 peptides to methylate DNA at heterochromatin. However, some concerns and suggestions are listed below.

We thank the reviewer for the assessment and have addressed your concerns detailed below.

1. The major concern is that the cryo-EM structures cannot explain how DIM2 is strictly regulated by HP1 and H3K9me3. The biochemical assays show that DIM2-mediated DNA methylation is strictly controlled by HP1 and H3K9me3 peptide, and DIM2 requires the presence of both H3K9me3 and HP1 proteins for a substantial DNA methylation activity (Figure 1b). However, the cryo-EM structure of DIM2 is highly similar to those ones in the DIM2-HP1 and DIM2-HP1-H3K9me3-DNA complexes with only minor differences in the TRD loop (disorder to order), RFTS domain (a loop reposition slightly for DNA interaction), and the bridging loop. This minor differences cannot explain the strict control of DIM2 by HP1 and H3K9me3. The authors need to consider other possibilities, such as that the cryo-EM structures of DIM2 alone and DIM2-HP1 complex are likely in the “activated” but not in the “auto-inhibitory” conformation. The previous studies in mammalian DNMT1 demonstrated that RFTS domain in DNMT1 is flexible with two conformations, activated and auto-inhibitory conformations, and the H3 binding induces the RFTS domain to shift from auto-inhibitory to activated conformation. In other words, the molecular mechanism of the interplay between DIM2, HP1 and H3K9me3 is likely not fully revealed by this study.

We thank the reviewer for the comments. We wish to clarify the following observations in this study. First, the structural disorder of the TRD region in apo-DIM2 is highly extensive, involving 58 residues that account for over one-third of the entire TRD. Such an extensive structural disorder in the TRD is unprecedented among the C5-DNA methyltransferases reported to date and presumably gives rise to a high energy barrier for the activation of DIM2. For instance, both the TRD and catalytic loop of DNMT1 are well defined in DNA-free state (see Supplementary Fig. 14f). Second, we wish to clarify that the RFTS domain of DIM2 shows a consistent, open conformation among all the classified groups in our cryo-EM data processing, suggesting it represents a dominant conformation. In fact, such an open conformation of the RFTS domain is consistent with the structural model predicted by AlphaFold (Fig. R1a). Third, the DIM2 and DNMT1 RFTS domains possess distinct surface charge distribution: DNMT1 presents an acidic patch to block the DNA-binding surface of the MTase domain (Fig. R1a); in contrast, the

electrostatic potential for the corresponding region in DIM2 RFTS domain is rather neutral, supporting the structural observation that DIM2 and DNMT1 RFTS domains are involved in distinct inter-domain interactions. Together, these observations testify that the HP1 binding-induced disorder-to-order transition of the DIM2 TRD controls the activity of DIM2, rather than an autoinhibitory regulation by the RFTS domain.

Following the reviewer's comment, we have clarified these points (Line 213 and Line 409-411 and Supplementary Fig. 14d,e) in the revised manuscript.

Figure R1. Structural analysis of the DIM2 RFTS domain. (a) Structural overlay of the cryo-EM structure (this study) and AlphaFold model of apo-DIM2. (b,c) Electrostatic surface of DNMT1 (b) and DIM2 (c) RFTS domains. The MTase-contact site of DNMT1 RFTS and its corresponding region in DIM2 RFTS domain are marked by circled by dotted lines.

2. The DIM2 mutation of R1104A at the TRD loop reduced the methyltransferase activity of DIM2 by 8-fold compared to the wild-type DIM2 (Figure 3h). It is concluded that “This observation therefore supports the notion that HP1 allosterically regulates the DNA methylation activity of DIM2 via structural stabilization of the DIM2 TRD”. It is actually not known if R1104A mutation could reduce or enhance the stability of DIM2 TRD loop. The assertion that HP1 binding enhances the stability of the TRD loop, consequently increasing DIM2 activity, lacks validity.

To address the reviewer's concern, we performed the thermal shift assay for DIM2, WT or R1104A mutant, HP1, and a mixture of DIM2 and HP1. As shown in Fig. R2, WT and R1104A DIM2 alone both yielded a T_m of 39.5 °C, suggesting that the TRD-residing R1104A mutation does not affect the stability of the TRD in apo-DIM2, in line with the structural disorder of the TRD in apo-DIM2. Meanwhile, HP1 alone (black) in two-fold molar excess than that of DIM2 (green and purple) yielded a much-reduced fluorescence signal, with a T_m of 58.5 °C. On the other hand, the co-incubation of WT DIM2 with HP1 led to a 5.0 °C higher T_m than that of WT DIM2 alone, consistent

with the formation of the DIM2-HP1 complex that further stabilizes the TRD. In contrast, the presence of HP1 only leads to a 2.5 °C increase of T_m for the R1104A mutant, supporting that the R1104A mutation, through disruption of the interaction between the DIM2 TRD and HP1, reduces the stability the TRD in the DIM2-HP1 complex. We have included this data as Supplementary Fig. 7 in the revised manuscript.

Figure R2. Thermal shift assay for wild type (WT) and R1104A-mutated DIM2, free or in presence of HP1. (a,b) Raw fluorescence data (a) and first derivative of the raw data (b). The difference in melting temperature (T_m) between free and complexed WT or R1104A DIM2 is indicated, respectively.

3. The ITC binding results are strange. Introducing the W261A mutation in the RFTS domain led a ~2.3-fold enhanced binding affinity (Fig. 6c and page 16). This result does not support the cryo-EM structure showing that the methylated K9 in the H3 peptide interacts with W261 in the RFTS domain stated in page 16. Is there other evidence to support that the H3 peptide is bound in the RFTS domain of DIM2?

As discussed in the manuscript, we attribute the “enhanced” H3K9me3-binding affinity of W261A-mutated DIM2 to the fact that the apparent K_d measured for WT reflects the collective contribution of two independent binding events by the RFTS and BAH1 domains, which makes the determination of the K_d using one-site binding model inaccurate. We indeed have additional ITC evidence supporting the RFTS-H3K9me3 binding, as shown in Fig. R3.

Figure R3. ITC binding curve of DNMT1 RFTS domain with the H3K9me3 peptide. We have included the data as Supplementary Fig. 12d in the revised manuscript.

4. The H3 peptide binding to DIM2 may be affected by the presence of DNA. To clearly elucidate the interplay among DIM2, H3 peptide and DNA, it is suggested to measure the affinity between DIM2 (WT verse mutants) and H3 peptide in the presence of DNA, or measure the DNA-binding affinity of DIM2 (WT verse mutants) in the presence of the H3 peptide.

Following the reviewer's suggestion, we have performed the EMSA experiment to compare the DNA-binding affinity between WT and mutant DIM2, in the presence of H3K9me3 peptide. As shown in Fig. R4, both the W261A and W581A mutations led to reduced DNA binding for DIM2, supporting the role of the RFTS-H3K9me3 and BAH1-H3K9me3 interactions in facilitating the substrate association of DIM2. We thank the reviewers for the suggestion and have included this data as Supplementary Fig. 12f in the revised manuscript.

Figure R4. EMSA analysis of the interaction between DIM2 and a DNA duplex in the presence of the H3K9me3 peptide. Each sample contains 1 pmol of DNA, 0 (in the absence of DIM2) or 8 pmol (in the presence of DIM2) of H3K9me3 peptide, and various amounts of DIM2 protein. The bands corresponding to free DNA and the protein-DNA complex are indicated.

Minor points:

1. The interactions between DIM2 and HP1 in the Interface I shown in Figure 2c are too busy to reveal any information.

Thank you for the suggestion. Reviewer 3 also pointed out this. We have modified Figure 2c to clarify the presentation in the revised manuscript.

2. Figure 8 is not referred in the main text.

Figure 8 was cited in the discussion section of the main text (Line 513).

Response to Reviewer 3

Reviewer #3 (Remarks to the Author):

I co-reviewed this manuscript with one of the reviewers who provided the listed reports. This is part of the Nature Communications initiative to facilitate training in peer review and to provide appropriate recognition for Early Career Researchers who co-review manuscripts

We thank the reviewer for his/her effort in reviewing the manuscript.

Response to Reviewer 4

Reviewer #4 (Remarks to the Author):

DNA methylation and histone H3K9me3 play pivotal roles in heterochromatin formation, which is essential for gene regulation and genome stability. However, the molecular mechanisms underlying the crosstalk between DNA methylation and repressive histone modifications are largely unknown.

In this study, Shao et al. focused on the structural basis of H3K9me3-binding dependent DNA methylation of DIM2 in *Neurospora*, which belongs to the superfamily of mammalian DNMT1. DIM2 consists of an N-terminal tail (NT), RFTS, BAH1, BAH2, and methyltransferase (MTase) domains. DIM2-dependent DNA methylation and heterochromatin formation are required for HP1 binding, which is an H3K9me3 reader protein. To reveal the molecular basis for DIM2 activation by HP1 and H3K9me3, the authors determined the cryo-EM structures of apo-DIM2, the DIM2-HP1 complex, and the DIM2-HP1-H3K9me3 complex. I am very impressed with the many findings of this study; direct HP1-binding contributes to the structural formation of the TRD of the MTase domain in DIM2. The DNMT1 RFTS domain binds to two-monoubiquitinated histone H3, which is essential for the recruitment of DNMT1 to the methylation site and DNMT1-activation. In contrast, the RFTS and BAH1 domains of DIM2 simultaneously bind to H3K9me3, which is required for the activation of DIM2. This different regulation of DIM2 by H3K9me3 is fundamental to the regulation of DIM2 activation by the repressive histone mark. Overall, this study clearly revealed H3K9me3 and HP1 binding dependent DIM2 regulation. These results are solid and include the novelty of the regulation of DNA methylation. This paper deserves to be published in Nature Communications. Before publication, the authors are encouraged to improve the messy figures, especially Figures 2c, 5b, 6a, and 6b, to make them clearer and easier to understand.

We thank the reviewer for his/her positive assessment of our study and support for publication of this manuscript. We also apologize for the lack of clarity regarding some of the figures. Following the reviewer's suggestion, we have modified Figures 2c, 5b, 6a and 6b to clarify the presentation.

REVIEWERS' COMMENTS

Reviewer #2 (Remarks to the Author):

The authors have addressed my concerns. The revised manuscript clearly reveals how DIM2 interacts with HP1 and histone H3K9me3 peptides to methylate DNA at heterochromatin.

Reviewer #2 (Remarks to the Author):

The authors have addressed my concerns. The revised manuscript clearly reveals how DIM2 interacts with HP1 and histone H3K9me3 peptides to methylate DNA at heterochromatin.

We thank the reviewer for his/her positive assessment of our manuscript.